# MCM2-7 ring closure involves the Mcm5 C-terminus and triggers Mcm4 ATP hydrolysis

Sarah V. Faull [1,6], Marta Barbon [1,2,6], Audrey Mossler [1], Zuanning Yuan [3], Lin Bai[3,4], L. Maximilian Reuter [1,5], Alberto Riera[1], Christian Winkler[1], Indiana Magdalou[1], Matthew Peach [1], Huilin Li [3,7] ✉ & Christian Speck [1,2,7] ✉

The eukaryotic helicase MCM2-7, is loaded by ORC, Cdc6 and Cdt1 as a double-hexamer onto replication origins. The insertion of DNA into the helicase leads to partial MCM2-7 ring closure, while ATP hydrolysis is essential for consecutive steps in pre-replicative complex (pre-RC) assembly. Currently it is unknown how MCM2-7 ring closure and ATP-hydrolysis are controlled. A cryo-EM structure of an ORC-Cdc6-Cdt1-MCM2-7 intermediate shows a remodelled, fully-closed Mcm2/Mcm5 interface. The Mcm5 C-terminus (C5) contacts Orc3 and specifically recognises this closed ring. Interestingly, we found that normal helicase loading triggers Mcm4 ATP-hydrolysis, which in turn leads to reorganisation of the MCM2-7 complex and Cdt1 release. However, defective MCM2-7 ring closure, due to mutations at the Mcm2/Mcm5 interface, leads to MCM2-7 ring splitting and complex disassembly. As such we identify Mcm4 as the key ATPase in regulating pre-RC formation. Crucially, a stable Mcm2/Mcm5 interface is essential for productive ATP-hydrolysis-dependent remodelling of the helicase.

The accurate replication of DNA is of extraordinary importance for genome stability. Eukaryotic cells have evolved regulatory mechanisms which separate helicase loading in late M/G1 phase from helicase activation in S-phase. Consequentially, this system guarantees that DNA replication can occur only once during the cell cycle[1,2] and that helicase loading is extremely efficient, as a defective loading reaction cannot be corrected later[3–6]. In eukaryotes, the ring-shaped heterohexameric protein, MCM2-7, represents the core of the replicative helicase. MCM2-7 loading, also known as pre-replicative complex (pre-RC) formation, is a multi-step process involving the binding of the Origin Recognition Complex (ORC) to DNA, followed by recruitment of Cdc6[7,8]. The ORC/Cdc6 complex recruits Cdt1/MCM2-7 to form an ORC/Cdc6/Cdt1/MCM2-7 (OCCM) intermediate[9]. The OCCM is a

transient complex and can be observed only in the presence of non- or slowly hydrolysable ATP analogues, such as ATPγS. In contrast, ATPase mutations in the Orc, Cdc6 and Mcm proteins fail to arrest complex assembly at the OCCM stage[10–12], indicating that the central ATPase during pre-RC formation is not known. Cryo-EM structures have revealed that double-stranded DNA (dsDNA) becomes inserted into the MCM2-7 ring during OCCM formation, which in turn induces partial ring closure[9,13]. Upon ATP-hydrolysis, thought to originate from Orc1 and MCM2-7[10–12], Cdc6 and Cdt1 are rapidly released[14]. ORC then flips from the C-terminal to the N-terminal side of MCM2-7 to form the MCM2-7/ORC (MO) complex, which consequently supports head-to-head loading of a second MCM2-7[12,14–18]. The head-to-head orientation facilitates the generation of bidirectional replication forks when

[1]DNA Replication Group, Institute of Clinical Science, Imperial College London, London, UK. [2]MRC London Institute of Medical Sciences, London, UK. [3]Structural Biology Program, Van Andel Research Institute, Grand Rapids, MI, USA. [4]Present address: Department of Biophysics, School of Basic Medical Sciences, Peking University, Beijing, China. [5]Present address: Institute of Molecular Biology (IMB) gGmbH, Mainz, Germany. [6]These authors contributed equally: Sarah V. Faull, Marta Barbon. [7]These authors jointly supervised this work: Huilin Li, Christian Speck. ✉e-mail: Huilin.Li@vai.org; chris.speck@imperial.ac.uk

the MCM2-7 double-hexamer becomes activated during the G1-S transition[19].

Within the pre-RC complex, Orc1-5, Cdc6, Cdt1 and Mcm3-Mcm7 all contain C-terminal winged-helix domains (WHDs), which are well known to participate in protein-protein and protein-DNA interactions[20]. The C-termini of Mcm3-Mcm7 consist of a flexible linker and a WHD, while Mcm2 has only a very short 20-residue extension (Fig. 1a). During OCCM formation, the Mcm3 C-terminus makes the first contact between ORC/Cdc6 and Cdt1/MCM2-7, while the WHD of Cdt1 functions to remodel the Mcm6 C-terminus to allow MCM2-7 to interact with ORC/Cdc6[6,12,21,22]. In complexes lacking the Mcm6 C-terminus, the OCCM arrests at the stage where the Mcm3 and Mcm7 C-termini are in contact with ORC, but Mcm6 and Mcm4 WHDs are not tethered, and DNA is not inserted into the MCM2-7 channel[23]. The C-terminal extension of Mcm5 (C5), which consists of a linker and a WHD, was not resolved in the OCCM structure[13] and therefore its specific role in pre-RC formation is unknown, especially as it has been shown that C5 is not required for Cdt1/MCM2-7 association with ORC/Cdc6[22].

Work by several groups has started to reveal defined functions for ATP-hydrolysis in the pre-RC assembly pathway and provided structural snapshots of complex assembly[3–6,17,22]. However, little is known about the structural changes that occur during ring closure, what triggers the all-important ATP hydrolysis process, and which enzymes are involved. Here, we investigate the function of C5, finding it to be essential for pre-RC formation via biochemical dissection and for ring closure in a new cryo-EM structure of the OCCM. Our data reveals that C5 enables MCM2-7 to "sense" ring closure, by the formation of a closed Mcm2/Mcm5 gate and interactions between MCM2-7 and ORC. Point mutations in the C5-Orc3 interface have allowed us to identify that Mcm4 is the key ATPase in pre-RC formation. We show that mutations at the Mcm2/Mcm5 interface trigger Mcm4 ATP hydrolysis, which promotes complex disassembly via splitting of the MCM2-7 ring. However, in the context of the unchallenged reaction, Mcm4 ATPase activity triggers Cdt1 release and supports the progression of pre-RC formation. We therefore reveal a structural and regulatory concept which connects MCM2-7 ring closure to pre-RC ATP-hydrolysis.

## Results

### The Mcm5 C-terminus is essential for pre-RC formation

Previous biochemical and structural work have started to explain important roles of the Mcm3, Mcm4, Mcm6 and Mcm7 C-termini during pre-RC formation, but the role of the Mcm5 C-terminus is poorly understood[12,13,21,22]. To address the function of C5 in this process, we generated an MCM2-7 construct harbouring a deletion of residues 693-775, termed MCM2-7-ΔC5. Overexpression of MCM2-7-ΔC5 in yeast caused reduced growth (Fig. 1b), and in a plasmid shuffle assay mcm5ΔC failed to rescue viability (Fig. 1c); suggesting that C5 has

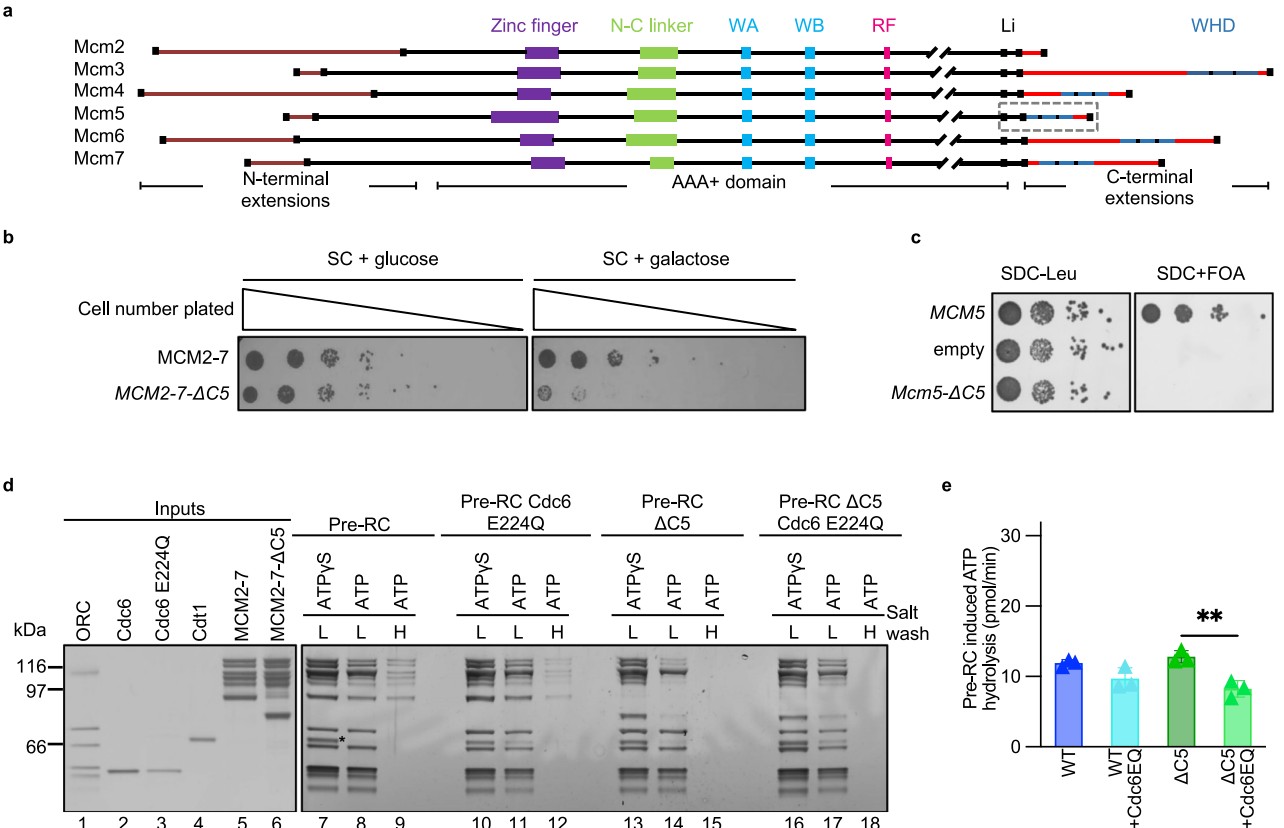

**Fig. 1 | The Mcm5 C-terminus is essential for MCM2-7 loading. a** Schematic illustration of the domain organisation of the *S. cerevisiae* MCM2-7 subunits. The domains include: Walker A (WA); Walker B (WB); arginine finger (RF); linker (Li) and winged helix domain (WHD). A dashed box highlights the Mcm5 C-terminus (C5). **b** Yeast strains were tested for dominant lethality following overexpression and by plating in a 5-fold dilution series on a dropout synthetic complete (SC) medium with either 2% glucose (no expression) or 2% galactose (expression). **c** Plasmid shuffle assay with a single copy plasmid containing the *MCM5* gene or *mcm5ΔC5* in the context of its natural promoter. A serial dilution of cells was plated on SDC–Leu or SDC + FOA plates. **d** Silver-stained gel of pre-RC reactions in the presence of ATPγS or ATP. Bead-coupled DNA, ORC, Cdc6, Cdt1 were assembled with WT MCM2-7 or MCM2-7-ΔC5 in the presence or absence of the Cdc6 E224Q ATPase mutant, subjected to low (L, 100 mM potassium glutamate) or high (H, 300 mM sodium chloride) salt washes. The asterisk (*) denotes the location of Cdt1 band that is retained in the presence of ATPγS. **e** Pre-RC induced ATPase activities of WT MCM2-7 and MCM2-7-ΔC5 in the presence of Cdc6 or Cdc6 E224Q (Cdc6EQ). Data are represented as the mean of three biological repeats and error bars represent the standard deviation. Data were analysed using unpaired, two-tailed t-tests. **P = 0.0058. Source data are provided as a Source Data file.

an important function in vivo. To address its role in pre-RC formation we performed an in vitro assay using purified proteins (Fig. 1d)[24]. In the presence of ATPγS, both wild-type (WT) MCM2-7 and MCM2-7-ΔC5 could assemble an OCCM (Fig. 1d, lanes 7 and 13), demonstrating that MCM2-7-ΔC5 has no defect in complex assembly. In the presence of ATP and WT MCM2-7, the pre-RC reaction proceeds rapidly. This results in the formation of reaction intermediates and eventually the MCM2-7 double-hexamer[17]. Upon ATP-hydrolysis, Cdt1 is rapidly released during pre-RC formation, therefore it is almost absent from low salt-washed pre-RC reactions (Fig. 1d, lane 8)[25]. In the high salt-washed sample, only the salt-resistant MCM2-7 double-hexamer remains bound to DNA (Fig. 1d, lane 9)[24,26]. However, in reactions containing MCM2-7-ΔC5 and ATP, we observed reduced complex formation after a low salt wash (Fig. 1d, lane 14). Moreover, high-salt stable double-hexamer formation was completely blocked (Fig. 1d, lane 15). This suggests that following OCCM formation, ATP-hydrolysis leads to disassembly of the MCM2-7-ΔC5 containing complexes and that any remaining low salt-stable complex is not functional for salt-stable double-hexamer formation.

We next performed pre-RC reactions using mixed populations of WT and mutant proteins in a 1:1 ratio and observed that WT MCM2-7 could not promote loading of the MCM2-7-ΔC5 complexes (Supplementary Fig. 1a, b). This result demonstrates that both hexamers require the presence of C5 for successful helicase loading. C5 is connected to the ATPase domain of Mcm5 via a 13 amino acid linker. To further validate the role of C5 and the linker, we generated mutant constructs that varied in either the linker or in the WHD itself (Supplementary Fig. 1c). A construct containing only the C5 linker, but not the WHD, was unable to form double-hexamers (residues 1-707, Supplementary Fig. 1d, lane 10), but a construct where the linker was replaced with a random "flexible linker" sequence could successfully load double-hexamers on DNA (Supplementary Fig. 1e, lane 10), confirming that the WHD of Mcm5 (WHD5), but not the linker, is essential for pre-RC formation. To address whether WHD5 can be functionally replaced with another WHD, we performed a pre-RC assay with mutants where WHD5 was replaced with either the WHD of Mcm4 (WHD4) or with a humanised sequence (hWHD5, 25% sequence conservation with *S. cerevisiae* at the level of amino acid identity), but the linker sequence from C5 was retained. The WHD4 failed to form double-hexamers (Supplementary Fig. 1f, lane 14), whilst the hWHD5 showed reduced formation compared to the WT protein (Supplementary Fig. 1f, lane 15). In summary, WHD5 is required for pre-RC formation, while the linker is not.

### The Cdc6 quality control stabilises MCM2-7-ΔC5 on DNA

As the MCM2-7-ΔC5-containing OCCM is rapidly destabilised upon ATP-hydrolysis (Fig. 1d), we sought to probe the underlying mechanism of the observed pre-RC instability. Two pre-RC ATP-hydrolysis pathways could be responsible: (1) ATP-hydrolysis by Cdc6 that has been implicated in quality control and results in the separation of Cdt1/MCM2-7 from ORC/Cdc6 or (2) ATP-hydrolysis by Orc1 or MCM2-7 that leads to the release of Cdc6 and Cdt1 from the pre-RC[12,14]. Initially, we used a Cdc6 E224Q ATPase mutant that has been shown to block pre-RC quality control, inhibits disassembly of defective pre-RC complexes[10,12,21] and displays strong dominant lethality effect in vivo[12,27]. In the pre-RC assay with WT MCM2-7, the addition of Cdc6 E224Q causes a mild stabilisation of Cdt1 (Fig. 1d, lane 8 vs 11) and a reduction in MCM2-7 double-hexamer formation when compared to reactions containing WT Cdc6 (Fig. 1d, lane 9 vs 12). This suggests that WT MCM2-7 only weakly activates the Cdc6 ATPase-dependent quality control pathway. With MCM2-7-ΔC5 we observed a Cdc6 E224Q dependent stabilisation of the low-salt washed complex (Fig. 1d, lane 14 vs 17), but whilst a block in Cdc6 ATP-hydrolysis stabilises the mutant complex, MCM2-7 double-hexamer formation still fails (Fig. 1d, lane 18). This indicates that in

the context of MCM2-7-ΔC5 Cdc6 ATP-hydrolysis promotes complex disassembly.

Next, we performed a pre-RC ATP-hydrolysis assay to quantify the effect of the Cdc6 E224Q mutant. Pre-RC ATPase activity originates from ORC, MCM2-7 and Cdc6 ATP-hydrolysis[10-12]. "Pre-RC-induced ATPase activity" is defined as the ATPase activity of the full pre-RC reaction minus the activity of the individual components (ORC/Cdc6 + Cdt1/MCM2-7) (Supplementary Figs. 2 and 15)[12]. In the presence of WT Cdc6, there was no notable difference in MCM2-7-ΔC5 pre-RC induced ATPase activity when compared to MCM2-7 (Fig. 1e). In the presence of the Cdc6 ATP-hydrolysis mutant, we observed a reduction in induced ATP-hydrolysis for both MCM2-7 and MCM2-7-ΔC5. In summary, we observe that the Cdc6 ATPase mutant stabilises MCM2-7-ΔC5 but does not rescue double-hexamer formation (Fig. 1d, lane 18) and does not block overall ATP-hydrolysis. We conclude that Cdc6 ATPase helps to disassemble MCM2-7-ΔC5, as we observed a stabilisation of the pre-RC complex under low-salt conditions with Cdc6 E224Q (Fig. 1d, lane 17). However, the data also highlight that another ATPase must be triggered during pre-RC formation, as robust ATP-hydrolysis and Cdt1 release were observed, even in the presence of Cdc6 E224Q. The removal of C5 triggered complex disassembly but did not reveal the molecular function of the domain. For this reason, we turned to a structural analysis to gain greater insight into the role of C5.

### An OCCM structure with a fully closed MCM2-7 ring

Our original OCCM structure[13] uncovered four out of the five Mcm C-termini in the OCCM, with C5 remaining elusive, and revealed a MCM2-7-loading intermediate in which the Mcm2/Mcm5 gate is not yet fully closed (we will refer to this as "open-gate OCCM"). Although the Mcm2/Mcm5 gate is topologically closed at this region because a domain-swapped α-helix from Mcm5 extends over and interacts with the C-terminal AAA+ domain of Mcm2; the MCM2-7 N-tier ring is open at the Mcm2/Mcm5 interface. We noticed that the electron densities near this interface were much weaker than the rest of the structure, indicative of multiple conformations[13]. In this study, we have utilised the advances in EM data processing technology that have occurred since the publication of our original OCCM structure[13]. By employing signal subtraction and focused 3D classification techniques on the MCM2-7 ring, as outlined in the Methods section and Supplementary Fig. 3, we have identified a conformation characterised by the full closure of the MCM2-7 ring. This closed-ring conformation accounts for approximately 10% of the total particle pool and yields a cryo-EM structure at an overall resolution of 6.1 Å; (Fig. 2, Supplementary Figs. 3 and 4, Supplementary Table 1). In the "closed-gate OCCM" complex, Mcm5 interacts directly with Mcm2, eliminating the 10−15 Å; wide gap between Mcm2 and Mcm5 that was previously observed (Fig. 2b). The structure reveals two important features: First, the C5 is now ordered and visible at the interface between ORC and MCM2-7 (Supplementary Fig. 5a), implicating the involvement of C5 in the loading process. Second, the ordered DNA density is longer, as it is better stabilised by the closed MCM2-7 ring (Fig. 2c) and can therefore been seen interacting with Mcm2 (Fig. 2d and Supplementary Fig. 5c). The details of these interactions are described below.

### The OCCM complex encloses an elongated stretch of DNA

In the closed-gate OCCM, we can observe a continuous 46 base pair (bp) dsDNA density entering the ORC-Cdc6 complex and emerging at the N-tier ring of MCM2-7. An additional 5−6 bp DNA was not modelled due to weaker densities near the N-tier exit (Fig. 2c). In the previous OCCM structure, the DNA density was shorter, either because the partially open MCM2-7 ring allowed access of DNaseI in that region, or because the DNA exhibits greater flexibility when not enclosed by the Mcm2/Mcm5 gate. The budding yeast replication origin, termed autonomously replicating sequence (ARS), contains an essential A element and two important B elements (B1 and B2). Using the ORC-DNA

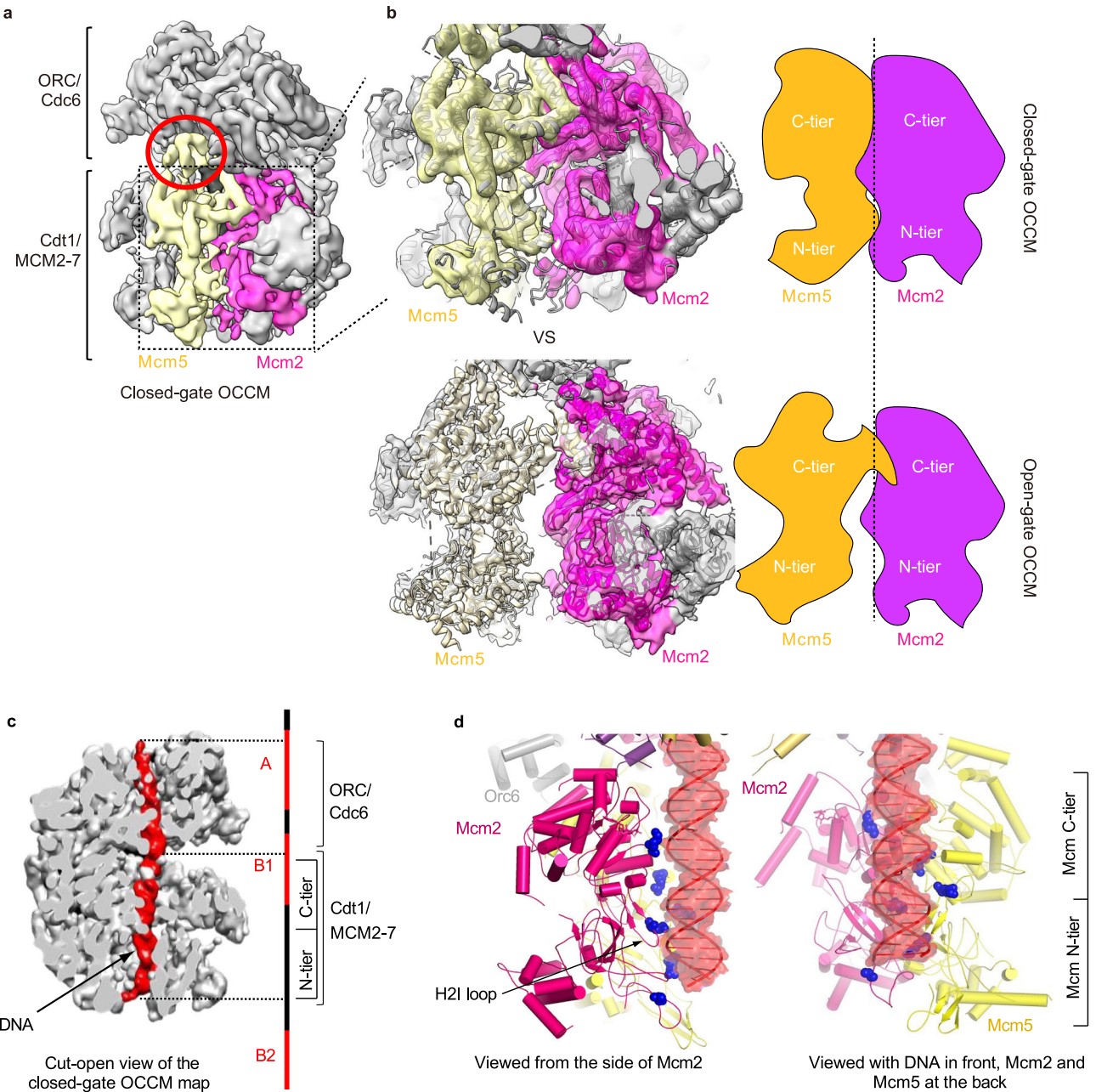

**Fig. 2 | Identification of a closed-gate OCCM structure. a** The cryo-EM 3D map of the closed-gate OCCM shown in surface rendering with the Mcm2 (magenta) and Mcm5 (yellow) subunits coloured individually. The C-terminal extension of Mcm5 (C5) is circled in red. **b** The topological conformations of Mcm2 and Mcm5 in the closed- and open- (PDB ID 5V8F) gate OCCM. Map-to-model fits and cartoon representations are shown to highlight the formation of interface between Mcm2 and Mcm5 in the closed-gate map. In the open-gate confirmation there is a gap of 10–15 Å between the subunits. **c** A segmented closed-gate OCCM with the DNA (in red) structurally aligned with the origin DNA sequence of ARS1 using the ORC-DNA structure (PDB ID 5ZR1) as reference. The locations of the A, B1 and B2 elements are denoted. **d** The protein-DNA contacts observed in the Mcm2/Mcm5 region of the closed-gate OCCM. Positively-charged residues, thought to electrostatically interact with DNA and facilitate ring closure, are shown in blue.

cryo-EM structure as a reference to align ORC on the ARS1 sequence[28], we suggest that in the OCCM complex ORC/Cdc6 protects the A element and part of B1, while Cdt1/MCM2-7 protects the latter half of the B1 element and nearly reaches up to the B2 element (Fig. 2c). Based on the higher resolution open-gate OCCM we have modelled the protein-DNA interface (Fig. 2d). In the N-terminal section of the MCM2-7, we observed that the helix-2 insertion loop of Mcm2 is in proximity with the phosphate backbone of the DNA, while Mcm5 is more distal to the DNA (Fig. 2d). Since open- and closed- gate OCCM contain identical protein complexes and only vary in their DNA length, the data implies that the full-length DNA contributes to MCM2-7 ring closure.

## MCM2-7 ring closure is associated with a reorganisation of the ORC/Cdc6-Cdt1/MCM2-7 interface

The MCM2-7 complex forms a toroidal shape, resembling a doughnut. Within this structure, C5 is connected by a flexible linker and can move inward to block the hole of the doughnut-shaped MCM2-7 assembly[29] (Supplementary Fig. 6). Interestingly, C5 is undetectable when MCM2-7 is loaded by ORC-Cdc6 to form the open-gate OCCM[13]. Although the side chain of the residues in Mcm5 WHD are not visible in our closed-gate OCCM structure, the previously determined structure of WHD5 assists[29,30] us in docking WHD5 into the cryo-EM map as a rigid body (Supplementary Fig. 5c). The stabilisation of C5 upon gate closure

reveals previously unobserved interactions between ORC and MCM2-7 (Supplementary Fig. 7a). C5 was observed next to Orc3 and the AAA+ domain of Mcm5 and we also noticed that the Orc2 N-terminal section is close to C5, consistent with a crosslinking analysis of the open-gate OCCM[13]. The Orc3-C5 interaction observed in the closed-gate OCCM is sterically hindered in the context of the open-gate OCCM complex, providing a structural explanation as to why C5 can only interact when the gate is fully closed (Supplementary Fig. 7b). Since the full MCM2-7 ring closure is dependent on complete DNA insertion and C5 only binds to the closed ring, the structural data suggests that C5 probes for successful DNA insertion and ring closure.

In the ORC-DNA and ORC-Cdc6-DNA complexes, Orc6 has been observed to interact with DNA intimately[28,31]. In the open-gate OCCM structure, both Mcm5 and Orc6 are flexible and disordered and visible predominantly at a low display threshold (Supplementary Fig. 8a). When the Mcm2/Mcm5 gate closes, as observed in the closed-gate OCCM structure, both Mcm5 and Orc6 are stabilised as their densities are visible even at a high display threshold (Supplementary Fig. 8b). Within the open-gate OCCM complex, C5 and Orc6 are positioned next to each other, and both interact with Orc3, but do not directly interact[32]. The closed-gate OCCM structure suggests that a central section of Orc6 bridges ORC and MCM2-7 through direct contact with the AAA+ domain of Mcm2. Unlike the other Mcm subunits that have a WHD (Fig. 1a), Mcm2 has a 20-residue C-terminal helical extension which is attached to its AAA+ section and appears to make specific contacts with Orc6 (Supplementary Fig. 9a). Surprisingly, the section of Orc6 that is in contact with DNA in the ORC-Cdc6-DNA complex[28,31] becomes restructured in the closed-gate OCCM complex and now forms contacts with Mcm2 (Supplementary Fig. 9a). Thus, in the closed-gate OCCM, ORC-DNA interactions are supressed. We hypothesise that this may be a mechanism to ensure that the DNA-loading reaction is unidirectional.

To test the biological relevance of the potential Orc6-Mcm2 interaction during pre-RC formation, we deleted the last C-terminal α-helix of Mcm2 (residues 848-868) generating MCM2-7-ΔC2. We chose to mutate Mcm2 over Orc6, as Orc6 contacts DNA and contributes to origin recognition[28] and therefore mutations in Orc6 may lead to ambiguous results. The overexpression of MCM2-7-ΔC2 resulted in dominant lethality (Supplementary Fig. 9b). In a pre-RC assay, we observed very efficient protein recruitment to the origin DNA in the presence of ATPγS (Supplementary Fig. 9c, lanes 6 and 7). In the presence of ATP, MCM2-7-ΔC2 resulted in mildly reduced (~65% compared WT) complex formation in low salt-washed reactions (Supplementary Fig. 9d, lane 6 vs 7, and 9e). Under high-salt conditions defective MCM2-7 double-hexamer formation was observed (Supplementary Fig. 9d, lane 8 vs 9). The biochemical data show that C2 plays a role in pre-RC formation. As we observed that MCM2-7-ΔC2 displayed reduced complex stability, we wondered whether this is due to the Cdc6 ATP-hydrolysis quality control pathway. Using the Cdc6 ATPase mutant (Cdc6 E224Q), we observed a slight stabilisation of MCM2-7-ΔC2 containing complexes in low salt-washed pre-RC reactions (Supplementary Fig. 9f). This suggests that Cdc6 ATP-hydrolysis may affect complex stability. Therefore, we performed ATPase assays and found that MCM2-7-ΔC2 displayed significantly elevated pre-RC ATP-hydrolysis (Supplementary Fig. 9g). The hydrolysis rate of MCM2-7-ΔC2 was not significantly reduced in the presence of Cdc6 E224Q (Supplementary Fig. 9g, green vs orange bars). This indicates that in the case of ΔC2, and similar to ΔC5, most of the ATP-hydrolysis is channelled through the MCM2-7 pathway and not the Cdc6 quality control mechanism.

## Mutations in the Mcm5 C-terminus induce ATP-hydrolysis

To further probe the role of C5 in pre-RC formation, we used point mutations that, in contrast to MCM2-7-ΔC5, would maintain the overall domain architecture of Mcm5. We generated three C5 amino acid substitution mutants in the context of full-length Mcm5 (Fig. 3a and Supplementary Fig. 10a). Due to the resolution of our model, we combined multiple point mutations in predicted helices as we cannot be certain of side-chain positions. We charge-swapped for maximal impact and focussed on charged or conserved amino acids (Supplementary Fig. 10b). Mcm5-O3 contains two mutations within the α1 and α2 helices of the WHD, targeting the upper tip of C5, near to the C5-Orc3 interface. Mcm5-AW and Mcm5-WH contain mutations in conserved amino acids of C5. The mutations of Mcm5-AW are in the two C-terminal α-helices at the WHD-AAA+ domain interface. Mcm5-WH contains mutations in the first two α-helices of the Mcm5-WHD near to the C5-Orc3 interface.

In vitro analysis showed that the three mutants are competent for efficient OCCM complex formation, which occurs in the absence of ATP-hydrolysis (Fig.3b, lanes 5–8). However, low salt-washed reactions containing ATP revealed that all three mutants caused a degree of complex instability (Fig. 3b, lanes 9–12 and Fig. 3c) and reduced high-salt resistant MCM2-7 double-hexamer formation (Fig. 3b, lanes 13–16). In vivo analysis of these mutants in a plasmid shuffle assay, which involves the removal of the WT copy of *MCM5*, showed growth inhibition (Fig. 3d), while only overexpression of Mcm5-O3 resulted in dominant lethality (Supplementary Fig. 10c). We next performed pre-RC ATPase assays. All three mutants had significantly elevated levels of ATP hydrolysis when compared to WT MCM2-7 (Fig. 3e) and MCM2-7-ΔC5 (Supplementary Fig. 10d), with Mcm5-WH producing a particularly large induction in ATP-hydrolysis. Thus, the data indicate that Mcm5 WHD mutants are associated with reduced cell growth in vivo, defective double-hexamer formation and significantly increased ATPase activity.

To ask whether Cdc6 ATPase plays a role in the reduced complex stability and increased ATP-hydrolysis, we employed the Cdc6 E224Q Walker B mutant. Whilst lower levels of ATP hydrolysis were observed when the mutants were assayed in the presence of Cdc6 E224Q compared to WT Cdc6, rates were still significantly elevated compared the WT MCM2-7 control (Supplementary Fig. 10e). Moreover, Cdc6 E224Q had no significant impact on stabilising the C5 point mutants in a pre-RC assay (Supplementary Fig. 10f). Thus, another ATPase than Cdc6 must be activated in the context of the C5 point mutants and be responsible for the majority of ATP-hydrolysis and complex disassembly. In the following we focused on identifying the missing ATPase.

## Mcm2 and Mcm5 ATPase mutants elevate pre-RC ATP-hydrolysis

Mcm ATP-hydrolysis is dependent on protein interactions. Neighbouring Mcm proteins form composite ATPase motifs, with one subunit providing both the Walker A (WA) ATP-binding motif and Walker B (WB) ATP-hydrolysis motif, while the other subunit provides an arginine finger (RF) which connects the subunits and promotes ATP-hydrolysis (Fig. 4a). This creates nucleotide binding pockets at the interface between adjacent subunits (Fig. 4b). Our structure shows that C5 forms interactions in the context of the closed-gate OCCM and the biochemical analysis indicates that interference with these interactions affects ATP-hydrolysis. Thus, the data suggest a connection between ATP-hydrolysis and MCM2-7 ring closure. One possibility is that closing of the Mcm2/Mcm5 gate activates ATP-hydrolysis at this interface. Thus, mutations affecting the Mcm5-WB and Mcm2-RF could completely block ATP-hydrolysis. Alternatively, another Mcm interface could be responsible for ATP-hydrolysis and instead the Mcm2 arginine finger mutation could destabilise the Mcm2/Mcm5 interface and result in increased ATP-hydrolysis, similar to the C5 mutations.

To test these hypotheses, we performed ATPase assays with Mcm2-RF and Mcm5-WB mutants. Here we observed that Mcm2-RF led to a significant induction of ATP-hydrolysis, while Mcm5-WB also displayed increased ATP-hydrolysis, although at a reduced level (Fig. 4c).

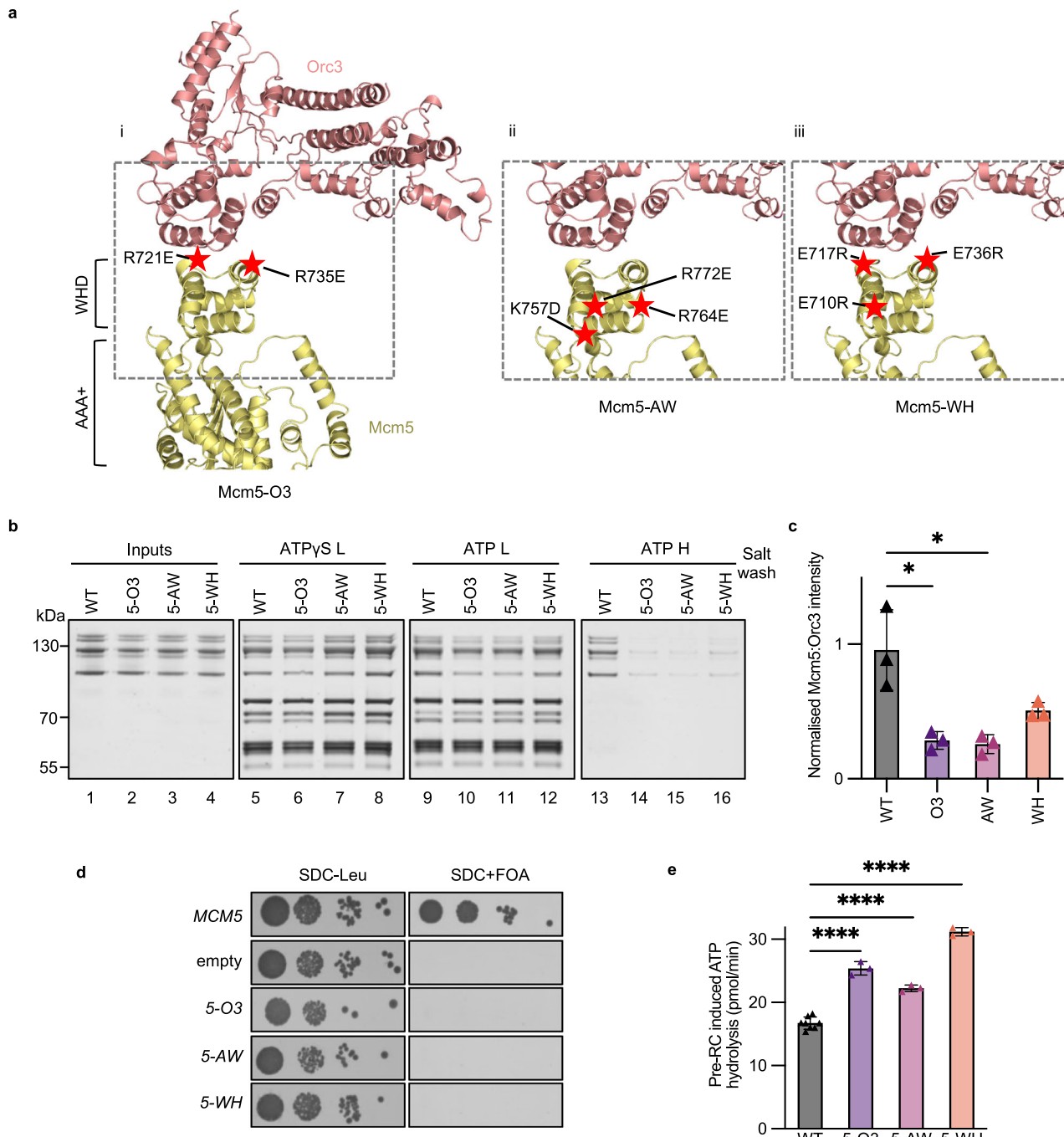

**Fig. 3 | Mutations in Mcm5 C-terminus induce pre-RC ATP-hydrolysis. a** The location of the mutated residues in the closed-gate OCCM Mcm5 C-terminus, as indicated by red stars. (i) Mutations at the Mcm5-Orc3 interface: Mcm5-O3, (ii) Mutations between the interface of the Mcm5 AAA+ and winged-helix (WHD) domains: Mcm5-AW (iii) Mutations across the Mcm5 WHD: Mcm5-WH. **b** Pre-RC assembly of WT MCM2-7 or Mcm5 mutant proteins. The reactions were assembled in ATP or ATPγS, washed with low (L) or high (H) salt and analysed by silver staining after elution from the beads using DNaseI. **c** Normalised Mcm5:Orc3 band intensity for WT MCM2-7 and MCM2-7 mutants for low salt washed reactions. Data are represented as the mean of three biological repeats, error bars represent standard deviation. Analysis was performed using a two-tailed t-test, *$P = 0.0190$ and $P = 0.0168$. **d** Plasmid shuffle assay with a single copy plasmid containing *MCM5* or mutants in the context of their natural promoters. A serial dilution series of cells was plated on SDC-Leu or SDC + FOA. **e** Pre-RC induced ATPase activities of Mcm5 mutants in the presence of WT Cdc6. Data are represented as the mean of three biological repeats for mutants and $n = 8$ for WT. Error bars represent standard deviation. Analysis was performed using a two-tailed t-test, ****$P = < 0.0001$. Source data are provided as a Source Data file.

In the pre-RC assay, the two mutants displayed normal OCCM formation in the presence of ATPγS (Fig. 4d, lanes 4–6), however complex assembly was particularly defective for Mcm2-RF in the presence of ATP (Fig. 4d, lanes 7–9). Consequently, Mcm2-RF could not load a salt-stable double-hexamer, while Mcm5-WB exhibited reduced loading compared to WT MCM2-7 (Fig. 4d, lanes 10–12). Even in the

presence of Cdc6 E224Q these mutants displayed significantly increased ATP-hydrolysis rates when compared to a WT MCM2-7 control (Supplementary Fig. 11a). Thus, the data show that the Mcm2/Mcm5 interface is not responsible for the bulk in ATPase activity. Instead, disruption of the Mcm2-RF motif resulted in a major increase in ATP hydrolysis.

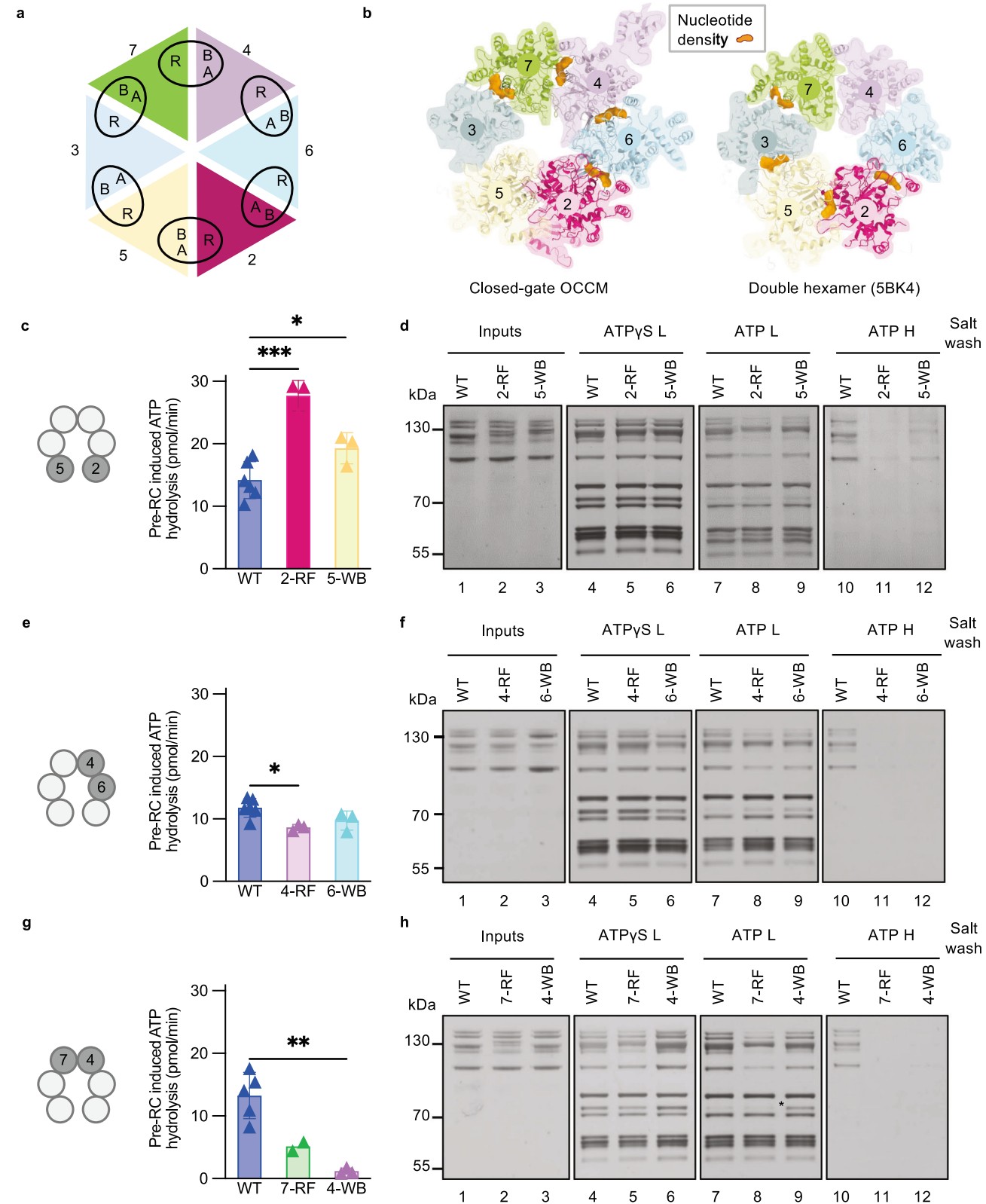

We wondered which ATPase could be involved in this process. In the closed-gate OCCM we observed four ATPγS molecules at the Mcm subunit interfaces of Mcm6/Mcm2, Mcm6/Mcm4, Mcm4/Mcm7, Mcm7/Mcm3, but no ATPγS density at the Mcm3/Mcm5 and Mcm5/Mcm2 interfaces (Fig. 4b). The same nucleotide occupancy was observed in the open-gate structure, however, in the next intermediate in the pre-RC pathway, the MO complex, where Cdc6 and Cdt1 have

been released; no nucleotide was observed at the Mcm6/Mcm4 and Mcm4/Mcm7 interfaces[17] and the same was observed in the double-hexamer[33] (Fig. 4b). Thus, the data indicate that ATP-hydrolysis could occur at Mcm6/Mcm4 and/or Mcm4/Mcm7 interfaces. To test the hypothesis, we generated the respective Mcm WB and RF mutants and measured their ATPase activities. Mutating the motifs at Mcm6/Mcm4 interface did not considerably alter ATPase hydrolysis rates (Fig. 4e)

**Fig. 4 | The Mcm4 Walker B motif is key for pre-RC induced ATP hydrolysis.**
**a** Cartoon showing the composite ATPase domain structure of MCM2-7. The Walker A and B motifs interact with the arginine finger (R) of the adjacent subunit.
**b** Schematic comparing the nucleotide occupancy (shown in orange) in the closed-gate OCCM and double hexamer (PDB ID 5BK4). Numbers correspond to organisation of Mcm subunits. **c** Pre-RC induced ATPase activities of WT MCM2-7, Mcm2 arginine finger mutant (Mcm2-RF) and Mcm5 Walker B mutant (Mcm5-WB). Means were calculated from three biological repeats for mutants and $n = 6$ for WT, error bars represent standard deviation. Data were analysed using a two-tailed t-test ***$P = 0.0003$, *$P = 0.0390$. **d** Pre-RC assembly of WT MCM2-7, Mcm2-RF and Mcm5-WB. The reactions (**d**, **f**, **h**) were assembled in ATP or ATPγS, washed with low (L) or high (H) salt and analysed by silver staining after elution from the beads using

DNaseI. **e** Pre-RC induced ATPase activities of WT MCM2-7, Mcm4 arginine finger mutant (Mcm4-RF) and Mcm6 Walker B mutant (Mcm6-WB). Means were calculated from three biological repeats for mutants and $n = 6$ for WT and error bars represent standard deviation. Data were analysed using a two-tailed t-test, *$P = 0.0109$. **f** Pre-RC assembly of WT MCM2-7, Mcm4-RF and Mcm6-WB. **g** Pre-RC induced ATPase activities of WT MCM2-7, Mcm7 arginine finger mutant (Mcm7-RF) and Mcm4 Walker B mutant (Mcm4-WB). Means were calculated from three biological repeats for Mcm4-WB, $n = 2$ for 7-RF and $n = 5$ for WT. Error bars represent standard deviation. Analysis was performed using a two-tailed t-test, **$P = 0.0016$ (**h**) Pre-RC assembly of WT MCM2-7, Mcm7-RF and Mcm4-WB. Source data are provided as a Source Data file.

and similar results were observed in the presence of Cdc6 E224Q (Supplementary Fig. 11b). In the pre-RC assay, both mutants displayed near normal OCCM complex formation in the presence of ATPγS (Fig. 4f, lanes 4–6), reduced complex assembly in the presence of ATP (Fig. 4f, lanes 7–9) and defective double-hexamer formation (Fig. 4f, lanes 10–12). Thus, we conclude that the Mcm6/Mcm4 interface is not blocking pre-RC ATP hydrolysis, and the Mcm6-WB/Mcm4-WB mutations destabilise pre-RC formation in an ATP-hydrolysis independent fashion.

Subsequently, we focussed on the Mcm4/Mcm7 interface. Mcm7-RF showed a marked reduction in ATP hydrolysis, and Mcm4-WB exhibited an even greater effect (Fig. 4g). Consistently, in the presence of Cdc6 E224Q, the ATP hydrolysis of Mcm4-WB was absent, while Mcm7-RF displayed a reduction, when compared with WT MCM2-7 (Supplementary Fig. 11c). In the pre-RC assay, Mcm7-RF showed stable OCCM formation with ATPγS, while Mcm4-WB was even more efficient than WT MCM2-7 in OCCM complex formation (Fig. 4h, lanes 4–6). In the presence of ATP, Mcm7-RF showed reduced MCM2-7 association (Fig. 4h, lane 8) and was unable to produce salt-stable MCM2-7 double-hexamer (Fig. 4h, lane 11). Mcm4-WB showed high levels of Cdt1 retention in the presence of ATP (Fig. 4h, lane 9) and blocked salt-stable double-hexamer formation (Fig. 4h, lane 12). In the presence of Cdc6 E224Q, the mutants exhibited significant reductions in pre-RC ATP hydrolysis when compared to WT MCM2-7 and the other ATPase mutants tested (Supplementary Fig. 11a–c). Thus, the data show that Mcm4 is the presiding ATPase during pre-RC formation and dominantly responsible for Cdt1 release.

To determine whether the mutations in the Mcm5 WHD indeed activate the Mcm4/Mcm7 ATPase, we made a double mutant. We combined Mcm4-WB and Mcm5-WH as these mutants showed the largest reduction and largest induction in ATPase hydrolysis, respectively. We termed this mutant Mcm4-WB/5-WH. Using this mutant, we observed that the pre-RC ATPase hydrolysis rate was greatly reduced (Supplementary Fig. 11d), similar as with Mcm4-WB (Fig. 4g). With the addition of Cdc6 E224Q, we were unable to detect any hydrolysis (Supplementary Fig. 11d). In a pre-RC assay, we observed stable complex formation with both WT Cdc6 and Cdc6 E224Q, as expected (Supplementary Fig. 11e). Thus, the data show that the increased ATP-hydrolysis by Mcm5-WH is dependent on the Mcm4/Mcm7 ATPase interface, while Cdc6 ATPase activity has a lesser role.

## Mcm4 ATP-hydrolysis leads to a structural change in MCM2-7 triggering Cdt1 release

Our data show that the Mcm4/Mcm7 interface is central for ATP-hydrolysis. We observed that in the presence of the Mcm4-WB mutant, Cdt1 is greatly stabilised. Thus, we suggest that Mcm4 ATP-hydrolysis triggers a structural change[5], which leads to Cdt1 release. To understand this in more detail we compared the structures of the closed-gate OCCM complex with the MO complex. The MO complex represents a post ATP-hydrolysis state, where Mcm4 ATP-hydrolysis has occurred, and the nucleotide has been ejected. When comparing Mcm4 of the closed-gate OCCM with the MO one can observe that N-terminal and

C-terminal domains rotating against each other (Supplementary Fig. 12a). This is highlights that Mcm4 ATP-hydrolysis leads to a major reorganisation of the Mcm4 subunit. In the OCCM, Cdt1 binds both the N-terminal and C-terminal domains of MCM2-7. When aligning the closed-gate OCCM and MO it becomes clear that the ATP-hydrolysis induces a structural change in Mcm4 and leads to a clash with Cdt1 (Supplementary Fig. 12b). As such our analysis now provides a structural explanation for ATP-hydrolysis dependent Cdt1 release.

## Failure of the MCM2-7 ring to close around DNA induces ring splitting at the Mcm4/Mcm7 interface

It is not known what happens to the MCM2-7 ring when helicase loading is defective, e.g., in the context of MCM2-7-ΔC5. We speculated whether during unproductive helicase loading Mcm4 ATP hydrolysis could potentially disrupt the MCM2-7 ring. This would imply that the Mcm4/Mcm7 interface can be broken in an ATP-hydrolysis-dependent fashion. We initially tested this concept in the context of purified MCM2-7 and the absence of other pre-RC proteins. We reasoned that stressing the MCM2-7 ring, e.g., by the addition of high salt, could lead to MCM2-7 ring breakage. To perform the experiment, we utilised the HA-tag on Mcm3, which is located in the middle of the Mcm3/5/7 trimer (Fig. 5a). We incubated MCM2-7 in solution and then coupled it to anti-HA beads, followed by low and high salt wash. Post wash, the proteins specifically bound to the beads were eluted using HA-peptide. Under low-salt conditions we were able to detect all six MCM2-7 subunits using SDS-PAGE (Fig. 5b) and by mass photometry (Fig. 5c). The trimeric peak detected by mass photometry is likely a by-product of the purification process or may indicate that MCM2-7 dynamically dissociates and reassociates in solution. After performing a high salt wash with WT MCM2-7, we were able to visualise bands for Mcm3/5/7 but not Mcm2/4/6 (Fig. 5b). Using mass photometry, we were no longer able to detect a mass corresponding to hexameric MCM2-7, but a dominant trimer peak (Fig. 5d, green histogram). Applying stress to the Mcm ring during a high salt wash therefore appears to destabilise the open MCM2-7 ring and induce splitting at the Mcm4/Mcm7 interface. To elucidate if this process is ATP-dependent, we repeated the experiments using the Mcm4-WB mutant. Under high-salt conditions, we were able to detect bands for all six Mcm subunits by SDS-PAGE (Fig. 5b) and a hexameric peak by mass-photometry (Fig. 5c, blue histogram). We note that after the high-salt wash both the trimer and hexamer peaks were reduced in context of Mcm4-WB. We suggest that this reduction was due to the high salt wash causing partial dissociation of the Mcm complex from the magnetic beads. Our results highlight that salt-induced stress results in MCM2-7 ring splitting at the Mcm4/Mcm7 interface, but preventing ATP hydrolysis at this interface stabilises the complex.

Next, we asked whether Mcm4-WB can also stabilise the ΔC5 and Mcm5-WH mutants, by comparing the Mcm5 mutants by themselves and in combination with Mcm4-WB (double mutants). We performed the Mcm3 pulldown under low-salt conditions and observed hexameric protein for all MCM2-7 complexes assayed (Fig. 5e). Under high-salt conditions, we observed "ring splitting", as indicated by the

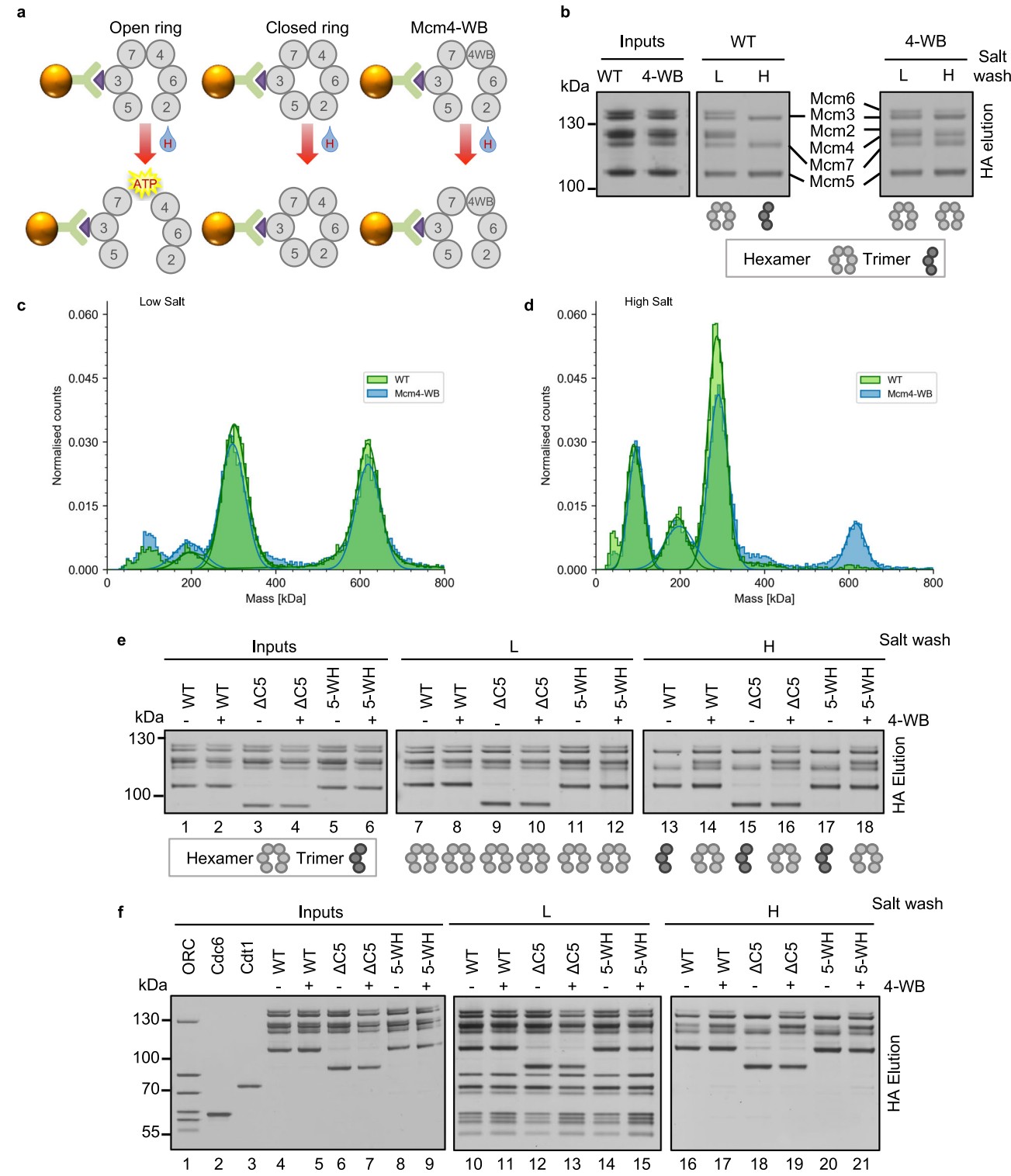

retention of the Mcm3/5/7 trimer, in WT MCM2-7, MCM2-7-ΔC5 and Mcm5-WH (Fig. 5e, lanes 13, 15 and 17). Introduction of the Mcm4-WB mutation in these complexes resulted in the ability to pulldown hexameric MCM2-7 after a high-salt wash, suggesting that ATP hydrolysis at the Mcm4/Mcm7 interface is key for ring splitting (Fig. 5e, lanes 14, 16 and 18).

## The Mcm4 Walker B mutant prevents ring splitting in complexes defective in ring closure

To ask whether MCM2-7 ring splitting occurs in the context of pre-RC formation and whether C5 is involved in this process, we adopted the

pulldown assay in the context of the full pre-RC. By adding ORC, Cdc6 and Cdt1 and a short piece of ARS1 DNA to the MCM2-7 complexes, we were able to pulldown all components under low-salt conditions (Fig. 5f, lanes 10-15). Since pre-RC formation is dependent on all proteins and DNA, the data suggest that normal pre-RC assembly occurred in the context of the Mcm3 pulldown assay. We note that for WT MCM2-7, under high-salt conditions, we observed bands of all Mcm subunits (Fig. 5f, lane 16) and the same was true with Mcm4-WB (Fig. 5f, lane 17). Next, we analysed the Mcm5 mutants. With MCM2-7-ΔC5 and Mcm5-WH we could detect pre-RC assembly under low-salt conditions (Fig. 5f, lane 12 and 14), but under high-salt conditions we could

**Fig. 5 | Mutating the Mcm4 Walker B motif stabilises the Mcm ring. a** Schematic showing the pulldown assay used to assess ring splitting. Magnetic beads (brown spheres) were used to pulldown MCM2-7 using the HA-tag on Mcm3. During a high salt wash (H), ATP hydrolysis results in splitting of the ring at the Mcm4/Mcm7 interface in open-ring MCM2-7 (left). Closed ring-MCM2-7, such as double hexamers (DH), are stabilised by closure of the Mcm2/Mcm5 gate. The Mcm4 Walker B mutant (Mcm4-WB) prevents ATP hydrolysis at the Mcm4/Mcm7 interface, providing ring stability. **b** Pre-RC pulldown assay: WT MCM2-7 or Mcm4-WB were incubated in solution before being pulled-down via Mcm3. Under low salt conditions (L), the proteins maintain their hexameric forms. After a high-salt wash (H), MCM2-7 splits into Mcm2/4/6 and Mcm3/5/7 subcomplexes, with only Mcm3/5/7 being specifically eluted from the HA beads. Mcm4-WB maintains hexameric integrity. **c** Refeyn mass photometry of pulldown samples, washed with low salt

buffer and then eluted using HA peptide. Trimer and hexamer peaks are visible for both WT MCM2-7 and Mcm4-WB. **d** Samples prepared as per part (**c**), but subjected to a high salt wash. Under these conditions, there is no detectable hexameric peak for WT MCM2-7. **e** Pulldown of Mcm proteins. After a high salt wash, ring splitting is observed in proteins lacking the Mcm4-WB mutation. **f** Pre-RC pulldown assay of proteins assembled on ARS1 DNA in solution. Under low salt conditions all Mcm proteins assemble pre-RC intermediates. After high salt washes hexameric MCM2-7 can be pulled down as a salt-stable DH. In complexes defective in ring closure, MCM2-7-ΔC5 and Mcm5-WH, trimeric protein is recovered. When combined with the Mcm4-WB mutation, the Mcm4/Mcm7 interface is stabilised, resulting in the elution of hexameric protein. Data are representative of three biological repeat and source data are provided as a Source Data file.

predominantly recover trimeric protein (Fig. 5f, lane 18 and 20). However, the addition of Mcm4-WB resulted in marked stabilisation and recovery of hexameric protein (Fig. 5f, lanes 19 and 21). This shows that integrating MCM2-7-ΔC5 and Mcm5-WH into the pre-RC is possible, but that MCM2-7 ring can split in an Mcm4/Mcm7 ATP-hydrolysis dependent fashion. We suggest that Mcm4 ATP-hydrolysis induced MCM2-7 ring splitting is a mechanism for removing pre-RC intermediates that encircle DNA, but fail to fully close the MCM2-7 ring.

Phosphorylation by CDK is another previously-characterised quality control mechanism that removes Cdt1-MCM2-7 from phosphorylated ORC, to ensure that helicase loading is blocked in S-phase[34]. We hypothesised that this complex disassembly could also work via MCM2-7 ring splitting at the Mcm4/Mcm7 interface. In our pulldown assay, ORC phosphorylation by CDK led to a phospho-shift of Orc2, which we could block by the addition of Sic1, a CDK inhibitor (Supplementary Fig. 13a, lanes 8 and 9). This phosphorylated ORC supported poor complex assembly with Cdc6, Cdt1 and MCM2-7 (Supplementary Fig. 13a, lane 11) and after the high-salt wash only trimeric MCM2-7 was retained (Supplementary Fig. 13a, lane 15). However, the addition of Mcm4-WB led to stable complex formation under low-salt conditions, even in the presence of phospho-ORC (Supplementary Fig. 13a, lane 13), and high salt-resistant MCM2-7 hexameric complexes (Supplementary Fig. 13a, lane 17). As such, the data suggest that S-phase CDK phosphorylated ORC also triggers Mcm4/Mcm7 ATP-hydrolysis dependent MCM2-7 ring splitting during pre-RC formation.

To compare the roles of Cdc6 and Mcm4 ATPase in removing loading-incompetent Mcm complexes, we performed a pre-RC assay using CDK-phosphorylated ORC along with relevant mutants (Mcm4-WB and Cdc6 E224Q) in the presence of ATP and ATPγS (Supplementary Fig. 13b). We used band densitometry (Supplementary Fig. 13c) to compare the effect of Cdc6 E224Q, Mcm4-WB and Cdc6 E224Q/Mcm4-WB on complex stabilisation. Considering that ORC phosphorylation impacts MCM2-7 recruitment, an early step in complex assembly[34,35], which Cdc6 ATPase controls[21,34,35], it was not surprising to see that Cdc6 ATP-hydrolysis had a significant impact on complex stability. This indicates that different quality control mechanisms exist that function more dominantly at the level of complex recruitment or DNA insertion. Based on the analysis of a series of mutants that interfere with MCM2-7 helicase loading, our data suggest that Mcm4 serves as the principal ATPase to remove unproductive helicase loading intermediates in a ring-splitting dependent fashion, but that it acts after the known Cdc6 quality control step, which removes predominantly early helicase loading intermediates before DNA insertion.

## Discussion

Here, we have established that the Mcm5-WHD functions for MCM2-7 ring closure. Our data show that interference with ring closure leads to induction of Mcm4 ATP hydrolysis and ring splitting. In contrast, we found that normal MCM2-7 ring closure leads to Mcm4 ATPase-

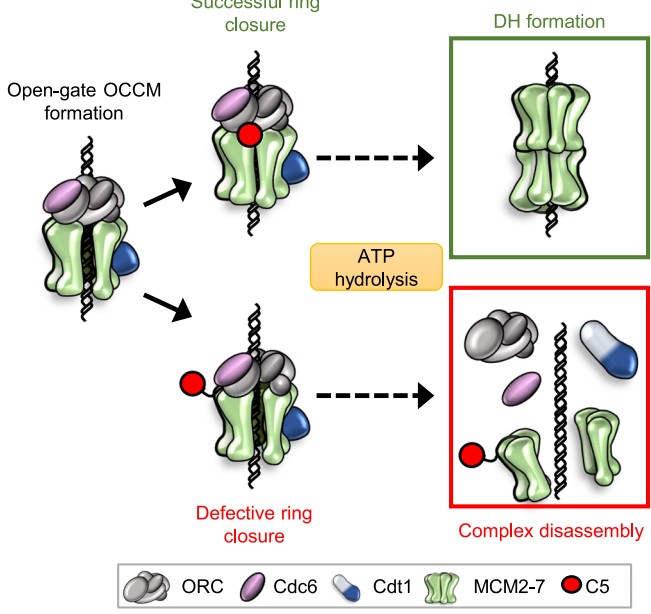

**Fig. 6 | The Mcm5 C-terminus is required for full closure of the Mcm gate to prevent ring splitting at the Mcm4/Mcm7 interface.** ORC, Cdc6, Cdt1 and MCM2-7 are assembled onto double-stranded DNA to form the open-gate OCCM. Engagement of the Mcm5 C-terminus (C5, red) leads to closure of the MCM2-7 ring, allowing the reaction to proceed to form a double hexamer via ATP hydrolysis – first by Cdc6 and then via Mcm4. This ATP hydrolysis facilitates first the release of Cdc6 and then Cdt1. If C5 is unable to engage, either due to truncation or disruptive mutations, then the ring cannot fully close. The Mcm ring therefore splits at the Mcm4/Mcm7 interface via ATP-hydrolysis. This leads to the dissociation of unproductive complexes and acts as a quality control mechanism.

dependent structural changes in the MCM2-7 ring, which promotes Cdt1 release in order to complete the pathway to double-hexamer formation (Fig. 6). Previous work had begun to determine the roles of the C-terminal extensions of the Mcm subunits, which make major contacts with ORC-Cdc6 during helicase loading[13]. Specifically, the C-termini of Mcm3 and Mcm6 have a role in ORC/Cdc6-dependent recruitment of Cdt1/MCM2-7[12,21,22]. Within the open-gate OCCM structure, only four out of five extensions were visible, and it was not clear whether C5 had a role in pre-RC formation[13]. Our closed-gate cryo-EM structure revealed that C5 becomes positioned at the interface of ORC and MCM2-7 once the MCM2-7 ring becomes closed. This structure also highlighted extended DNA density throughout the entire complex (Fig. 2c). We suggest that the presence of DNA itself acts as the driving force behind the MCM2-7 ring closure - the strong negative charge of the DNA pulls the positively charged inner channel of the Mcm subunits inwards. This leads to a reorganisation of the MCM2-7-DNA interface, i.e. the helix-2 insert of Mcm2, which forms a well-known helicase motif that is located in the central channel of Mcm proteins[36],

engages with the DNA (Fig. 2d). This is different from the DNA-MCM2-7 double-hexamer complex, where the helix-2 insert of Mcm2 adopts a more distal location, but similar to the single-stranded (ss) DNA-bound CMG[33,37–39]. In the context of the CMG, the Mcm2 helix-2 insertion may function in ATP-hydrolysis-driven translocation, whereas during pre-RC formation, the interaction between Mcm2 and DNA may contribute to MCM2-7 ring closure.

The Mcm5 C-terminus has previously been observed in MCM2-7 complexes not encircling DNA, such as in the Cdt1 bound MCM2-7 or in the apo CMG[29,40]. In these structures, C5 was found to be positioned near the central channel of the MCM2-7 ring, where DNA is usually located (Supplementary Fig. 6). In DNA bound complexes, such as the MCM2-7 double-hexamer with dsDNA, the MO with dsDNA[17], and the CMG with both ss and forked DNA[33,37–39]; C5 was not visible, suggesting that it adopts a flexible conformation. In all these structures C5 was not interacting with other proteins, and so its function remained elusive. Our work demonstrates that the deletion of C5 impacts cellular growth in budding yeast and that C5 plays an essential role in helicase loading (Fig. 1b–d). The closed-gate cryo-EM structure highlights that C5 adopts a fixed conformation, suggesting that the respective OCCM interaction surfaces only become available once the ring has closed (Supplementary Fig. 8b). Consistent with this hypothesis, C5 binding in the context of the open-gate OCCM cannot occur, as its binding surface is unavailable due to a steric clash between C5 and Orc3 (Supplementary Fig. 7b), thus providing a structural explanation for the ring closure induced re-localisation of C5. To test the biological relevance of the structural changes associated with MCM2-7 ring closure, we targeted a number of protein-protein interfaces by mutagenesis. C5-Orc3 (Fig. 3e), Mcm2-Mcm5 (Fig. 4c) or Mcm2-Orc6 (Supplementary Fig. 9g) interface-interaction mutations all activate pre-RC induced ATP-hydrolysis and negatively affect pre-RC formation and in vivo viability. Thus, our data suggest that suboptimal MCM2-7 ring closure triggers ATP-hydrolysis-dependent complex disassembly. Therefore, the data link ring closure with ATP-hydrolysis. We suggest that ring closure supports the correct alignment of the Mcm ATPase motifs, which in turn trigger MCM2-7 ATP-hydrolysis, similar as suggested in the context of the Cdc45-MCM2-7-GINS complex[41].

We note that the deletion of C5 drastically impacted complex stability but did not result in the further increase of pre-RC-induced ATPase activity that was observed with C5 point mutants. We suggest that C5 has an additional function in stabilising complex intermediates. Interestingly, the WHD of Mcm4 can bypass this severe complex instability of C5 deletion (compare Fig. 1d and Supplementary Fig. 1f). Thus, C5 may have a sequence-independent structural role. Considering that C5 sits within the central pore of the Cdt1-MCM2-7 complex[29,30] and the pre-insertion OCCM[23] (a complex arrested before DNA insertion), C5 may also function in stabilising the MCM2-7 ring during the initial stages of OCCM formation or regulate DNA insertion. Since these steps occur before ATP-hydrolysis, they may result in ATPase-independent instabilities and complex disassembly.

Following OCCM formation, ATP hydrolysis is induced, first promoting Cdc6 and then Cdt1 release[14]. We identified, through the use of ATPase mutants and ATPase activity measurements, two ATPases that function during pre-RC formation, namely Cdc6 and Mcm4. Considering that Cdc6 release occurs prior to Cdt1 release, and that Mcm4 ATPase is required for Cdt1 release, we suggest that Cdc6 ATP-hydrolysis is initiated first. What could trigger Cdc6 ATPase activity? It has been shown that correct ORC-Cdc6-DNA interactions are essential for complex stability and that mutations in the ARS sequence trigger Cdc6 ATP-hydrolysis[8,42]. As DNA straightening disrupts multiple ORC-Cdc6-DNA interactions, we suggest that DNA insertion triggers Cdc6 ATP-hydrolysis. This fits with single-molecule observations that DNA straightening is associated with Cdc6 release[16].

The role of MCM2-7 ATP-hydrolysis during pre-RC formation has been puzzling for many years[10–12]. Previous work identified that the

Mcm2, Mcm4 and Mcm6 WB motifs are essential for viability[10]. However, in vitro work was focussed on WA and RF mutations[10,11], which may impact complex stability. Based on the changes in nucleotide occupancy between the OCCM and MCM2-7 DH (Fig. 4b), we focussed on the WB mutants of Mcm4, Mcm5 and Mcm6 and corresponding RF mutations of Mcm7, Mcm2 and Mcm4. Importantly, the WB mutants had less impact on pre-RC stability and by combining data from corresponding interfaces with ATPase measurements, we could reveal that Mcm4 ATPase plays a central role in pre-RC ATP-hydrolysis. Mutations at the Mcm4/Mcm7 ATPase motif had the largest impact on pre-RC ATP-hydrolysis, and the Mcm4-WB mutant readily stabilised Cdt1 during complex formation. Thus, we conclude that the Mcm4/Mcm7 interface is the main source of MCM2-7 ATP-hydrolysis during pre-RC formation.

During ring-closure in the OCCM structure, only minor structural changes occur that are mainly localised to the N-tier ring of MCM2-7 (Supplementary Fig. 14a). In comparison, the structural changes that are observed between the closed-gate OCCM and the MO structure are much more drastic. These changes are likely to be the consequences of MCM2-7 ATP-hydrolysis, which is inhibited by the presence of ATPγS in the OCCM structure. A concerted shift in 5 out of 6 Mcm subunit N-terminal domains can be observed, relative to their C-terminal domains (Supplementary Fig. 14b). Since Cdt1 bridges the Mcm C- and N- terminal domains, the ATPase dependent structural changes impact on the Cdt1 binding sites, which leads to a misalignment of the sites and a clash between Cdt1 and the Mcm4 N-terminal domain (Supplementary Fig. 12b). Therefore, we suggest that these Mcm4 ATP-hydrolysis dependent structural changes promote Cdt1 release. It also appears possible that Cdc6 and Mcm4 ATPases synergise to promote MCM2-7 ring closing and Cdt1 release, since inhibition of the Mcm4 ATPase supressed Cdc6 ATP-hydrolysis. We suggest that Cdc6 and Mcm4 ATPase trigger release of Cdc6 from ORC and Cdt1 from MCM2-7, which in turn destabilises the OCCM. ORC can now reattach to the N-terminal face of the loaded single-hexamer.

Previously, Cdc6 ATP-hydrolysis, was shown to result in complex disassembly for pre-RC quality control[10]. Here we identify that Mcm4 ATPase activity has a significant role in this process. Crucially, we have demonstrated that Mcm4 can induce ATP-hydrolysis-dependent MCM2-7 disassembly in the context of defective ring closure (Fig. 6). Furthermore, we have determined this disassembly to be mediated by splitting of the MCM2-7 hexamer at the Mcm4/Mcm7 interface. A series of helicase loading mutants become readily stabilised in the pre-RC assay (Fig. Supplementary. 11e) and pulldown assay (Fig. 5e, f) when Mcm4 ATPase activity is blocked. Interestingly, ORC phosphorylation plays a key role in stopping helicase loading in S-phase and in this way stops re-replication and associated genomic instability[43]. Importantly, recent single-molecule and structural work have identified that phosphorylated ORC impacts in large pre-RC formation prior to DNA insertion[34,35]. In this context, we observed that Cdc6 ATPase had a larger impact than Mcm4 ATPase on the disassembly of ORC phosphorylated helicase loading intermediates (Supplementary Fig. 13b, c). As such, we feel that Cdc6 ATPase functions predominantly for disassembly of helicase intermediates prior to DNA insertion, while Mcm4 ATPase acts at the stage of DNA insertion. In summary, Mcm4, along with Cdc6, is a key regulator of pre-RC quality control.

Eukaryotic helicase loading can be distinguished from the bacterial version by the loading of two helicase complexes on the origin DNA. This enables the formation of bidirectional replication forks but also causes a major complication, as multiple structural intermediates are necessary to assemble two MCM2-7 hexamers in a single head-to-head double-hexamer[24,26]. Our data detail the key structural changes that reorganise the Mcm2/Mcm5 interface upon ring closure or clear the origin from failed helicase loading intermediates. Considering that several mutations at the Mcm2/Mcm5 gate affect complex stability and regulate Mcm4 ATP-hydrolysis, we hypothesise that these could be an

interesting drug target for developing a small molecule pre-RC inhibitor. Equally, blocking Mcm4 ATP-hydrolysis could drastically impact several functions of MCM2-7 in DNA replication. Crucially, cancer cells frequently lose the checkpoint that monitors successful helicase loading[44]. Therefore, inhibitors that affect human MCM2-7 ring closing could trick cancer cells into a situation where they fail to replicate their DNA and undergo apoptosis.

## Methods

### Cloning
A full list of the oligos, plasmids and strains used in this study is listed in Supplementary Data 1. *mcm2 (Δ848-868)* and *mcm5 (Δ693-775)* were created by site-directed mutagenesis of pCS12 (pESC-*Leu MCM2-MCM7*) or pCS232 (pESC-*URA* HA_*MCM3-MCM5*) yielding pCS491 and pCS495, respectively. All other mutant MCM2-7 constructs were commercially generated (Genscript). pET24a-Sic1 was constructed by cloning Sic1 amplified from the genomic DNA of the *S. cerevisiae* strain S288C[45] using the NheI/XhoI restriction endonuclease sites.

### ORC expression and purification
ORC was expressed and purified based on the previously described method[8]. Hi-5 cells (Invitrogen) were grown adherently and infected with baculoviruses (a kind gift from Bruce Stillman (CSHL)): Orc1,6 MOI 20; Orc2,5 MOI 15; Orc3,4, MOI 10) for 2 h before the addition of serum to the media. Protein was expressed for 48 h prior to cell collection via centrifugation (10 min, 4 °C, 270 × g). Pellets were washed with cold PBS and centrifuged again before resuspension in HYPO buffer (20 mM HEPES pH 7.5, 5 mM KCl, 1.5 mM MgCl$_2$, 1 mM DTT, 1x complete protease inhibitor (CIP) without EDTA (Roche)). Cells were lysed by douncing and nuclei collected by centrifugation (20 min, 4 °C, 25,000 × g, no brake). Nuclei were resuspended in H/0.2 buffer (50 mM HEPES pH 7.5, 0.2 M KCl, 5 mM magnesium acetate, 1 mM EDTA, 1 mM EGTA, 0.02% NP40, 1 mM DTT) with 1x CIP without EDTA and homogenised by douncing. ORC was extracted using ammonium sulphate precipitation (12.5%/45%). The 45% pellet was resuspended in H/0.2 buffer with 1x CIP without EDTA, 0.1 mM ATP, 5 mM MgCl$_2$, 14.3 mM β−Mercaptoethanol (H.0.2*). The above buffer without KCl (H/0*) was added before incubation for 20 min, with intermittent vortexing. Once dissolved, tubes were again centrifuged (15 min, 4 °C, 27,000 × g). Supernatant was loaded onto an SP Sepharose (20 ml HR16/60, resin from GE Healthcare) column and eluted with a gradient of 0.2–0.6 M KCl (H/0.2–H/0.6, with 14.3 mM β−Mercaptoethanol). Protein was diluted with 0.8 volume of H/0 and centrifuged (15 min, 4 °C, 27,000 × g). Supernatant was loaded onto a Mono Q 5/50 GL (GE Healthcare) and eluted using a gradient of H/0.2* – H/0.5*. Pooled fractions were diluted with H/0* with 1x CIP without EDTA and incubated overnight with SP-Sepharose resin. ORC was eluted from the resin using H/0.6* with 1x CIP without EDTA and loaded onto a Superdex 200 16/60 column (GE Healthcare) equilibrated with H/0.2 with 14.3 mM β−Mercaptoethanol. Protein was concentrated by binding to SP-Sepharose overnight and eluted with H/0.6 with 14.3 mM β−Mercaptoethanol. H/0.2 buffer was used to dilute to the required protein concentration (-1.6 mg/ml) and glycerol was added to a final concentration of 10%.

### Cdc6 (Cdc5 E224Q) expression and purification
Cdc6 was expressed and purified based on the previously described method[8], with minor modifications. *E. coli* BL21 codon-plus cells (Stratagene) were transformed with pGex-6P1-CDC6 (Amersham). Cultures were induced at OD600 0.9 with 0.5 mM IPTG for 5 h at 16 °C with shaking. The cells were lysed using lysozyme and sonicated in sonication buffer (500 mM potassium phosphate (K$_x$PO$_4$) pH 7.6, 150 mM potassium glutamate (KGlu), 5 mM MgCl$_2$, 1 mM ATP, 1 mM DTT, 1% (v/v) Triton X-100) with 2x CIP without EDTA and centrifuged at 21,000 × g at 4 °C for 30 min. The supernatant was

incubated for 2 h with glutathione-agarose (Sigma) on a rotating wheel. Beads were rinsed with sonication buffer, washed for 10 min, washed for a further 30 min, before being rinsed 3 times with cleavage buffer (100 mM K$_x$PO$_4$ pH 7.6, 150 mM KGlu, 5 mM MgCl$_2$, 1 mM ATP, 1 mM DTT, 1% (v/v) Triton X-100). Cleavage buffer containing 1:50 (v/v) PreScission Protease (Amersham) was added to the beads for 2 h. The eluted protein was collected and pooled with 3x rinses using wash buffer (50 mM K$_x$PO$_4$ pH 7.6, 75 mM KGlu, 5 mM MgCl$_2$, 1 mM ATP, 1 mM DTT, 0.1% (v/v) Triton X-100). An equal volume of dilution buffer (5 mM MgCl$_2$, 1 mM ATP, 1 mM DTT) was added along with 50% slurry of Hydroxyapatite beads and incubated on a rotating wheel for 30 min. Beads were washed with 2x wash buffer, followed by 2x with rinse buffer (50 mM K$_x$PO$_4$ pH 7.6, 150 mM KGlu, 5 mM MgCl$_2$, 1 mM DTT, 0.1% (v/v) Triton X-100, 15% (v/v glycerol), added drop-by-drop. Cdc6 was eluted using 50 mM K$_x$PO$_4$ pH 7.6, 400 mM KGlu, 5 mM MgCl$_2$, 1 mM DTT, 0.1% (v/v) Triton X-100, 10% (v/v) glycerol. For protein to be used in pre-RC conditions containing ATP, 6 mM ATP was added and incubated overnight at 4 °C on a rotating wheel.

### Cdt1 expression and purification
Cdt1 was expressed and purified based on the previously described method[24]. *E. coli* BL21 codon-plus RIL cells (Stratagene) were transformed with GST-Cdt1. Cultures were induced at OD600 0.9 with 0.5 mM IPTG for 5 h at 16 °C with shaking. The pellets were resuspended in sonication buffer (50 mM Tris-HCl pH 7.2, 150 mM NaCl, 5 mM MgCl$_2$, 1% Triton (v/v) X-100, 1 mM DTT and 1x CIP with EDTA) and sonicated in the presence of lysozyme. Supernatant was incubated with glutathione-agarose (Sigma) at 4 °C for 2 h. Beads were rinsed with sonication buffer, washed 2x with high salt buffer (50 mM Tris-HCl pH 7.2, 300 mM NaCl, 5 mM MgCl$_2$, 1 mM DTT, 1% (v/v) Triton X-100 and 2x CIP with EDTA), then twice washed and rinsed with sonication buffer without CIP. The GST tag was cleaved by addition of 1:50 (v/v) PreScission Protease (Amersham) for 2.5 h at 4 °C. Flowthrough was collected and pooled with the eluate of 3x rinses with wash buffer (25 mM Tris-HCl pH 7.2, 25 mM HEPES KOH pH 6.0, 75 mM NaCl, 5 mM MgCl$_2$, 1 mM DTT). 1 volume of dilution buffer was added (25 mM HEPES KOH pH 6.0, 5 mM MgCl$_2$, 1 mM DTT) prior to the addition of SP-Sepharose beads for 30 mins, 4 °C. Cdt1 was eluted drop-by-drop with elution buffer (25 mM Tris-HCl pH 7.2, 25 mM HEPES KOH pH 6.0, 250 mM NaCl, 5 mM MgCl$_2$, 1 mM DTT, 0.1% (v/v) Triton-X100). Protein was diluted to 0.5 mg/ml using elution buffer supplemented with 15% (v/v) glycerol.

### MCM2-7 purification
The WT and mutant MCM2-7 proteins were expressed and purified based on the previously described method[24]. Cells (OD600 0.5) were grown in SCGL medium supplemented with His overnight and then an equal volume of rich media was added the following morning. After 3 h, protein expression was induced by addition of 2% galactose for 3 h (OD600 1.8). Cultures were arrested with alpha factor (2.5 µg/mL) for 3 h. Cells were broken using a SPEX™ freezer-mill. The lysed cells were resuspended in buffer (25 mM HEPES KOH pH 7.5, 10 mM magnesium acetate, 50 µM zinc acetate, 10% (w/v) glycerol, 0.1% (w/v) Triton-X100, 50 µM EDTA, 300 mM KGlu, 3 mM ATP) with 1xCIP without EDTA and extracted for two hours at 4 °C with mixing. The cells were then centrifuged for one hour at 21,000 × g at 4 °C and the extract (supernatant) incubated with anti-HA affinity beads resin (EZview™ Red Anti-HA Affinity Gel, Sigma) at 4 °C for two h on a rotating wheel. To elute the protein, the beads were incubated with HA peptide (0.5 mg/ml) at 16 °C for 3 h and the eluted material was concentrated using a 15 ml Amicon Ultra 100 K centrifugal filter (Millipore), prior to injection to a gel filtration column (Increase Superose 6 10/300 (GE Healthcare)) to remove the contaminating Mcm3/Mcm5 dimer.

## Sic1 expression and purification

Plasmid pCS316 was transformed into *E. coli* BL21 CodonPlus(DE3)-RIL (Agilent) and expressed in terrific broth by addition of 0.2 mM IPTG at an OD = 0.5 for 2 h at 30 °C. Sic1 was purified as described by Brocca et al.[46]. Briefly, the pellet was resuspended in lysis buffer (950 mM sodium phosphate pH 8, 10 mM imidazole, 300 mM NaCl, with 1x CIP without EDTA) and heat treated for 10 min at 99 °C, followed by 10 min on ice and pelleted at 11,000 × $g$ at 4 °C for 10 min. The extract was then incubated with TALON beads (Clontech France) at 4 °C for 1 h on a rotating wheel. The protein was eluted with 50 mM sodium phosphate pH 8, 250 mM imidazole, 300 mM NaCl, with 1x CIP without EDTA and desalted with a PD10 column (GE Healthcare) pre-equilibrated with equilibration buffer (50 mM sodium phosphate pH 8, 200 mM NaCl). The elution was incubated with CM-sepharose (GE Healthcare) for 30 min and with 50 mM sodium phosphate pH 8, 600 mM NaCl.

## Pre-RC assay

The pre-RC was performed as previously described with minor modifications[24]. Briefly, 40 nM ORC, 80 nM Cdc6 (or Cdc6 E224Q), 40 nM Cdt1, 40 nM MCM2-7 (WT or mutant) were incubated on ice for 20 min in 50 µl pre-RC buffer (50 mM HEPES-KOH pH 7.5, 100 mM KGlu, 10 mM magnesium acetate, 50 µM zinc acetate, 5 mM DTT, 0.1% (v/v) Triton X-100, 10% (v/v) Glycerol, 3 mM ATP or ATPγS), followed by a 20 min incubated at 24 °C. 6 nM of pUC19-ARS1 DNA coupled to magnetic beads were added to the proteins and incubated for 20 min at 24 °C. Beads were washed 3x with either pre-RC buffer or high salt buffer (pre-RC buffer plus 300 mM NaCl) before the complexes were eluted from the beads using DNaseI. For ATPγS, reactions were washed with 1 mM nucleotide. Final products were separated by SDS-PAGE and analysed by silver staining. For quantification of low salt reactions, band intensities for Mcm5 and Orc3 were measured using Multi Gauge V2.3 (FujiFilm). All intensities were corrected against a background control. A ratio of Mcm5 to Orc3 was calculated and data from three independent experiments were normalised, plotted and analysed using an unpaired, two-tailed t-test in GraphPad Prism v9. For western blotting an anti-Mcm5 antibody[47] was used at a 1:5000 dilution.

## CDK Pre-RC assay

For the assay with CDK-phosphorylated ORC, all steps were performed at 24 °C with mixing at 1050 RPM in an Eppendorf Thermomixer. Firstly, 40 nM ORC was incubated with 40 nM CDK for 20 min in pre-RC buffer with 3 mM ATP. 120 nM Sic1 was then added for 10 min to inhibit CDK. 6 nM of pUC19-ARS1 DNA coupled to magnetic beads and ORC allowed to bind for 20 min prior to three washes with either 3 mM ATP or 1 mM ATPγS. 50 µl pre-RC buffer with 3 mM ATP or ATPγS was then added, followed by 80 nM Cdc6 (or Cdc6 E224Q), 40 nM Cdt1, 40 nM MCM2-7 (WT or mutant) for 20 min. Beads were washed 3x with either pre-RC buffer or pre-RC with 1 mM ATPγS before the complexes were eluted from the beads using DNaseI.

## ATPase assay

The ATPase assay was performed as previously described[12,42]. A 150 bp fragment containing the ARS1 sequence was amplified by PCR with forward primer (CAAAATAGCAAATTTCGTCAAAAATGC) and reverse primer (TTTACATCTTGTTATTTTACAGATTTTATGTTTAGATC) from the template plasmid pUC19 ARS1 (pCS372). 2.5 pmol of DNA was incubated with 2.5 pmol of MCM2-7 (or mutants), ORC, Cdc6 (or Cdc6 E224Q) and Cdt1 in 12.5 µl of buffer (25 mM HEPES, pH 7.6, 100 mM KGlu, 5 mM magnesium acetate, 1 mM DTT, 1 mM EDTA, 1 mM EGTA, 0.1% (v/v) Triton X-100, 10% (w/v) glycerol, 1 mM ATP) for 30 min on ice. 5 µCi of [α-32P] ATP (3000 Ci/mmol) were added to each reaction and incubated at 24 °C. After 60 min 2 µl were taken from each sample and incubated with 0.5 µl of stop solution (2% SDS, 50 mM EDTA). 1 µl sample was spotted on a thin layer chromatography cellulose plate and developed in 1 M formic acid and 0.5 M lithium chloride for 20 min.

Plates were analysed using Multi Gauge V2.3 (FujiFilm). Averages were calculated from at least three independent experiments, or two for Mcm7-RF and error bars represent the standard deviation. Data were analysed by unpaired two-way t-tests using GraphPad Prism v9 and statistically significant comparisons are denoted.

## Dominant lethality assay

A yeast strain was plated on a synthetic complete (SC) agar dropout plate (4% glucose) and incubated at 30 °C for 48 h, followed by an overnight incubation at 30 °C with agitation in SC dropout medium containing lactic acid and glycerol. A five-fold dilution series was performed and spotted onto plates containing glucose (non-inducing condition) or galactose (inducing condition) and incubated at 30 °C for the appropriate time (24–72 h).

## Yeast plasmid shuffle assay

Yeast strain and plasmid generation were performed as previously described with the following modification[48]. Briefly, a single copy of *MCM5* was knocked-out (*Δmcm5::TRP1*) in a diploid WT yeast strain (W303, YC162), which was subsequently transformed with as single-copy, centromeric plasmid encoding the *MCM5* ORF, 500 bp flanking region upstream, 400 bp downstream and the *URA3* locus (pRS316-*MCM5*). *TRP1*[+] and *URA*[+] yeast cells were sporulated and tetrads dissected. Retrieved spores, the shuffle strain, were tested for growth on selective media for confirmation. *mcm5* mutants were cloned into pRS315[49] using *AscI* and *XmaI* and sequence-verified plasmids were transformed into the shuffle strain. Transformants, grown on media lacking leucine, were dot spotted on synthetic media lacking leucine or plates containing 5-Fluoroorotic acid to shuffle out the *URA3*-containing plasmid at 30 °C for 3 days and analysed.

## HA-bead pulldown assay

150 nM MCM2-7 (WT or mutant) was incubated for 30 minutes at 24 °C, 500 RPM in 20 µl of pre-RC buffer. 6 µl of anti-HA magnetic beads (Pierce #88836) that had been equilibrated in pre-RC buffer were then added for 30 min, with mixing increased to 1050 RPM. The tube was applied to a magnetic rack and the supernatant removed prior to washing the bead 3x with 100 µl pre-RC buffer or pre-RC buffer with 300 mM NaCl for high salt washes. Protein specifically bound to the beads was eluted by the addition of 2 mg/ml HA peptide in 10 µl pre-RC buffer for 10 min at 24 °C, 1050 RPM. Final products were separated by SDS-PAGE and analysed by silver staining. For pre-RC pulldowns, 150 nM ORC, 300 nM Cdc6, 150 nM Cdt1, 150 nM MCM2-7 (WT or mutant) were incubated with 200 nM 290 bp fragment containing the ARS1 sequence amplified by PCR with forward primer (AACAGCTATGACCATG) and reverse primer (GTAAAACGACGGC-CAGT) from the template plasmid pUC19 ARS1 (pCS372).

## CDK phosphorylation pulldown assay

100 nM ORC was incubated for 20 min at 24 °C, 500 RPM in 20 µl of pre-RC buffer with or without 25 nM CDK. 50 nM Sic1 was then added to reactions containing CDK to prevent the phosphorylation of Cdc6 and MCM2-7 for 10 min at 24 °C, 500 RPM. 200 nM Cdc6, 75 nM Cdt1, 75 nM MCM2-7 (WT or mutant), 200 nM 290 bp DNA were then added for 30 min at 24 °C, 500 RPM. 4 µl of anti-HA magnetic beads (Pierce #88836) that had been equilibrated in pre-RC buffer were then added for 30 min, with mixing increased to 1050 RPM. Washes and elution were performed as described above. For the CDK and Sic1 control, CDK & Sic1 were pre-incubated together for 10 min at 24 °C, 500 RPM, prior to the addition of ORC.

## Mass photometry

The pulldown assay was performed as described above, with minor modifications: the concentration of MCM2-7 (WT or mutant) was increased to 300 nM and the pre-RC buffer contained 0.025% (v/v) Triton

X-100 and 5% (v/v) glycerol. Measurements were performed at room temperature using a Refeyn TwoMP Mass Photometer (Refeyn Ltd). Samples that had been eluted from HA-beads were diluted 1 in 10 on the instrument in Mass Photometry Buffer (25 mM HEPES pH 7.5, 250 mM potassium acetate, 4 mM magnesium acetate, 1 mM ATP, 1 mM DTT) and added to a silicone gasket composed of six wells (GraceBioLabs) on a pre-cleaned sample coverslip (Refeyn Ltd). AcquireMP software (v. 2023R2, Refeyn Ltd) was used to record movies for 60 s in the regular field of view. Analysis was performed using DiscoverMP software (v. 2023R2, Refeyn Ltd). Prior to recording the measurements, the instrument was calibrated using BSA (Sigma, 66 kDa monomer, 132 kDa dimer) and Bovine Thyroglobulin (Sigma, 670 kDa) in Mass Photometry Buffer to produce a linear mass calibration. The maximum mass error accepted for each calibration was below 5%, as defined by the DiscoverMP software. Measurements were completed in triplicate, with one representative histogram shown for each condition.

### Focused 3D refinement and model building of the "closed-gate OCCM" conformation

In our previous work reporting the 3.91 Å; structure of the OCCM, we collected approximately 7500 raw movie micrographs on a Gatan K2 direct electron detector in a Titan Krios microscope operated at 300 kV. After 2D and 3D classifications, we obtained a final dataset of 304,288 particles and the reported 3D density map[13]. In this follow-up study, we hypothesised that the weak densities in the Mcm2/Mcm5 gate region of the partially-open OCCM is likely due to the presence of multiple conformations. To test this hypothesis, we performed a "focused" 3D classification procedure in RELION 2.0[50]. We first used a soft-edged mask to generate an EM density map of the ORC-Cdc6 complex. We then used the ORC-Cdc6 map to subtract the ORC-Cdc6 signal from the experimental OCCM particles, leaving only the MCM2-7 signals in the subtracted raw particles. 3D classification was performed on the subtracted, MCM2-7 only particles without performing further image alignment. In addition to the previously-identified conformation in which the MCM2-7 C-tier ring is closed, but the N-tier ring is open, we found a conformation in which the Mcm2/Mcm5 gate was closed at both the C-tier region and the N-tier region, i.e., the MCM2-7 ring is fully closed. Further refinement of this map resulted in a final 3D map with an estimated resolution of 6.1 Å;. To build the atomic model, we started with the updated model of the open-gate OCCM structure (PDB ID 5V8F). The structures of individual proteins were first docked as rigid bodies into the 3D density map in Chimera v1.17[51] and Coot v0.9.8[52]. The smallest ORC subunit, Orc6, largely absent in the open-gate OCCM model, but had clear densities in the closed-gate conformation. We generated a homologue initial model from the crystal structure of the human Orc6 complex (PDB ID 3M03) using the online SWISS-MODEL server (https://swissmodel.expasy.org/). The initial model thus obtained was manually rebuilt in Coot. Bulky residues such as Phe, Tyr, Trp, and Arg – some resolved at this resolution – were used for amino acid sequence registration. The manually-built model was then iteratively refined in real space by phenix.real_space_refine[53] and then manually adjusted in Coot. We also performed a reciprocal space refinement procedure with the application of secondary structure and stereochemical constraints in the programme Phenix v1.14[54]. The structure factors (including phases) were calculated by Fourier transform of the experimental density map with the programme Phenix.map_to_structure_factors. The atomic model was validated using MolProbity v4.4[55]. Structural figures were prepared in UCSF Chimera v1.17, Chimera X v1.7 and The PyMOL Molecular Graphics System, Version 2.0 Schrödinger, LLC. Cryo-EM data collection and refinement statistics can be found in Supplementary Table 1.

### Reporting summary

Further information on research design is available in the Nature Portfolio Reporting Summary linked to this article.

## Data availability

The 3D cryo-EM map of the OCCM complex in the "closed-gate" conformation at 6.1 Å; resolution has been deposited in the EMDB database with accession code EMD-44441. The corresponding atomic model was deposited in the RCSB PDB bank with accession code 9BCX. Requests for resources and reagents should be directed to and will be fulfilled by the corresponding authors. Source data are provided with this paper.

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

## Acknowledgements

We thank all members of the Speck lab for critical reading of the manuscript. We thank Juergen Zech for the expression and purification of Sic1 and Silvia Tognetti for the expression and purification of CDK. We thank Bruce Stillman (CSHL) for the kind gift of the anti-Mcm5 antibody and ORC and CDK baculoviruses. C.S. was funded by the Biotechnology and Biological Sciences Research Council (BB/N000323/1 and BB/S001387/1), the Medical Research Council (MC_U120085811), the Wellcome Trust (107903/Z/15/Z) and Cancer Research UK (DRCNPG-May21\100006). L.M.R. was supported the Deutsche Forschungsgemeinschaft (DFG, German Research Foundation, project ID 505087959). H.L. was funded by the US National Institutes of Health (GM131754).

## Author contributions

S.V.F. expressed and purified MCM2-7 proteins and mutants, developed the Mcm4 ATPase hypothesis, generated the ATPase mutants and performed their analysis in pre-RC assays, developed the ring-splitting assay performed the analysis of ATPase data, generated conceptual and several EM figures. M.B. expressed and purified the C5 and C2 mutants, generated biochemical and dominant lethal data for C2 and C5 and initial ATPase assays. A.M. performed MCM2-7 expression and purification and final ATPase assays. A.R. generated the OCCM complex for the cryo-EM study. Z.Y. performed cryo-EM and 3D reconstruction, Z.Y., L.B., H.L. and C.S. analysed the cryo-EM data and produced EM figures. L.M.R. produced deletion strains and performed the plasmid shuffle assay. C.W.

performed the cloning of MCM2-7-ΔC2 and MCM2-7-ΔC5. M.P. performed mass-photometry experiments and analysis. I.M. and S.F. purified the ORC, Cdc6 and Cdt1 proteins. S.F., M.B. and C.S. designed and C.S. supervised the research. S.F., M.B., H.L. and C.S. wrote the manuscript.

## Competing interests

The authors declare no competing interests.
