## [Transparent Peer Review file · Nature Communications]

MCM2-7 ring closure involves the Mcm5 C-terminus and triggers Mcm4 ATP hydrolysis

Corresponding Author: Professor Christian Speck

Version 0:

Reviewer comments:

Reviewer #1

(Remarks to the Author)

Reviewer's comments on the manuscript 'MCM2-7 ring closure involves the Mcm5 C-terminus and triggers MCM2-7 ATP hydrolysis' by Li, Speck and coworkers.

The formation of the pre-replication complex involves ORC/Cdc6-mediated loading of Cdt1/MCM2-7 onto DNA origins. Significant advancements, particularly through cryo-EM structural analyses and in vitro assays using purified proteins by various research groups, including the Li-Speck groups, have greatly enhanced our comprehension of these complexes. Specifically, we now have a comprehensive understanding of how the OCCM intermediate forms and how this transient complex is detectable in the presence of ATP γ S. Subsequent to ATP hydrolysis, the release of Cdc6 and Cdt1, along with ORC flipping, leads to the establishment of MCM2-7-ORC (MO), which subsequently loads the second MCM in a head-to-head configuration. However, there are still pertinent inquiries regarding the structural alterations during MCM ring closure and the stimuli for ATP hydrolysis. This work aims to elucidate these critical questions.

Using elegant biochemical and structural approach, the authors demonstrate that the C-terminus of Mcm5 is essential for preRC formation and Mcm4 ATP-hydrolysis induces a structural change in the MCM2-7 complex triggering the release of Cdt1. The authors use advances in cryo-EM data analysis to reveal a remodeled "closed conformation" structure of the OCCM, that provides new insights into the structural changes and regulatory mechanisms behind MCM2-7 ring closure. Further, the authors provide compelling data related to the key roles played by Mcm4 and the mechanism of Mcm2-7 ring closure leading to Mcm4 ATP hydrolysis and ring splitting. These are important observations and fill a significant gap in the existing literature. I recommend the following to improve the presentation of the data.

1) One of the key observations is the importance of Mcm4 ATP-hydrolysis. I recommend including that in the title.
2) Some of the data using the invitro purified proteins and their interpretations are based on visual appearance of the complexes in high or low salt. There should be a better way to quantify this data. For example: In the presence of ATP, Mcm2-7-delC2 resulted in mildly reduced complex formation in low salt-washed reactions (Supplementary Fig 6D, lanes 6-7). This data is not compelling and authors need to provide some method of quantification. Similarly, for Figure 3c However, low salt-washed reactions containing ATP revealed that all three mutants caused a degree of complex instability. This data is not convincing. Also, Thus, the data indicate that Mcm5 WHD mutants are associated with reduced viability, defective.... There is no statistical significance included for the datasets. Another example, is Supplementary Figure 7C, with Mcm5-O3 showing the least effect and Mcm5-WH displaying the greatest stabilization.' This reviewer is unable to visualize this observation. Can you quantify this?

3) Fig 1:

D. In the lanes showing pre-RC C 5, we see that Cdt1 is not completely released/marginal levels of Cdt1 remains, which is slightly higher than its pre-RC WT counterpart after low salt extraction. In the later sections, we do see the direct role of the ATPase activity of MCM4 in Cdt1 release, which is downstream of the role of Mcm C5. Could loss of Mcm5 C5 activity by itself result in the reduction of Cdt1 release?

4) Fig 2:-The authors hypothesize that electrostatic interactions between DNA and MCM2-7 contribute to ring closure but do not experimentally verify these interactions. This interpretation is speculative without direct experimental evidence. As a follow-up, they could mutate the positively charged residues on Mcm2 and Mcm5 and measure how stable the OCCM is using assays like EMSA.

-Closed ring conformation only accounts for 10% of the particle pool suggesting that this conformation might not be

representative-however, given the dynamic and transient nature of the OCCM it might not be possible to capture the model at this specific conformation.

-Resolution of closed ring model is 6.1A which is relatively low

5) Fig 3:

C. In the absence of ATP hydrolysis, the mutations 5-AW and 5-WH show higher OCCM stability compared to WT. Is this because we find higher levels of ATP hydrolysis (hence higher OCCM disassembly) in the presence of these mutations compared to WT, so absence of ATP hydrolysis stabilizes the OCCM more than it stabilizes WT?

6) For Supplementary Figure 8a: Can the ATP hydrolysis be shown like the data presented in Figure 4c. Instead of normalizing the data, please show the WT and this would be consistent with how the data is plotted in the entire manuscript. Also, for none of the ATP hydrolysis experiments, statistical significance is calculated. Each experiment must be repeated at least thrice and the results plotted with statistics. This will improve the rigor of the presented data.

7) Supplementary Fig 9:

The results section titled "Mcm4 ATP-hydrolysis leads to a structural change in MCM2-7 triggering Cdt1 release" that discusses this figure would be easier to follow if the text includes references to the figure (specifying what section describes 9A or 9B). The text also talks about the comparison between the closed gate OCCM and MO complex structures, but the all the figures are labeled with "open gate OCCM" which is slightly misleading.

8) The various linkers for Mcm5 C-terminus were generated (Figure S1c). It would be useful to include a control for another MCM subunit for these experiments and test the addition of a linker to its C-terminus (of course guided by their structure).

9) Figure 4g is labeled Figure 5g in the text ...similar as with Mcm4-WB

10) The ORC phosphorylation experiments with the addition of CDK is interesting but doesn't tackle if a specific ORC is involved. This information didn't seem to fit well with rest of the theme of the manuscript.

11) Please fix typos-an example included, there are a few more: Cdt1 is binding both the N-termial and C-termianl domains of MCM2-7.

12) Incomplete sentences: To if this process is ATP-dependent, we repeated the experiments using the Mcm4-WB mutant.

13) Fig 5-Label for 5D is missing on the figure

Reviewer #2

(Remarks to the Author)

In this manuscript, Faull et al use a combination of cryo-EM, biochemistry and genetics to examine the role of various protein functions and domains during pre-RC assembly. Visualising a new intermediate in the MCM loading process, they suggest that the MCM5 winged helix domain closes the MCM2-7 ring in the OCCM complex, and that failure to do so triggers release of MCM2-7 in a manner that is dependent on the MCM4:MCM7 ATPase site. The focus of the manuscript is an interesting one; the text is well written and the experiments have been well executed. However, in my opinion there are some key aspects of the manuscript that currently preclude publication in Nature Communications.

Major points.

1. More detail of the processing steps involved in obtaining the new closed ring OCCM structure should be provided. In particular, a supplementary figure describing the processing steps, particle numbers involved etc should be added to the manuscript.
2. Related to this, while I can appreciate that the intermediate nature of the closed ring OCCM structure may limit the resolution that can be obtained, 6.1 A is likely to limit the conclusions that can be drawn from the structure. More detail is required around this. For example, there is no description at all in the text or figures of how well the density in their final map can be fitted to a model of the C-terminus of MCM5. Is the density unambiguously the MCM5 WHD or could another domain also be fitted here? This a crucial point. The same applies to the MCM2 residues shown in figure 2d – can sidechain densities for these residues be observed in the map? No fit of the model to the map is shown. If not, this should be removed.
3. While I appreciate that this may be challenging, no validation of the MCM5 WHD position, or description of any attempts to validate its position are provided. Cross linking mass spectrometry could be one way to do this.
4. They describe several point mutations in MCM5 that they state are at the interface of MCM5 WHD and either the AAA+ domain or the ORC3 interface. No evidence is shown for this. A figure showing the electron density map and model and the positions of the amino acids that are mutated should be shown. Again, how sure are the authors on the location of these residues given the resolution of the structure?
5. Perhaps related to the previous point, the biochemical impact of mutating the MCM5 WHD is different from deleting it. For example, MCM5 WHD deletion does not cause an increase in 'pre-RC induced ATP hydrolysis' (figure 1), whereas all mutations they study in the MCM5 WHD do (figure 3). This does not fit with their model that any failure to close the 2/5 gate triggers ATP hydrolysis by MCM4. How do the authors rationalise this? The MCM5 Δ WHD should be analysed alongside the MCM5 WHD point mutations in figure 3 as a point of reference.
6. Given that, as referenced in the paper, an ORC-Cdc6-dependent quality control mechanism was previously described by the Diffley lab, the authors should compare the role of Cdc6 and Mcm4 ATPase in removing loading-incompetent MCM complexes. Are they both required to remove the same loading-incompetent complexes?

Minor points.

1. There are a number of typos throughout the manuscript.

2. For the 'pre-RC induced ATP hydrolysis', the supplementary figure helps to understand where this figure derives from. However, the individual values going into this final figure are important and should be included in the main text/figures for each mutant analysed.

Reviewer #3

(Remarks to the Author)

Review of MCM2-7 ring closure involves the Mcm5 C-terminus and triggers MCM2-7 ATP hydrolysis by Faull et al. for Nature Communications. In this manuscript, the authors have added to our knowledge of MCM2-7 loading and activation by revealing three further aspects of this process. 1) They show that the winged helix domain within the CTE of MCM5 is influential in interacting with ORC3 to stabilize a closed OCCM complex, 2) that the ATPase activity of MCM4 is necessary for closing the MCM2-5 interface prior to C5 movement, and that 3) Cdt1 is ejected during this process. They also suggest that any defect in MCM2-7 ring closure would disassemble the entire complex with MCM4 ATPase activity providing a failsafe for properly loaded and closed MCM2-7 complexes. The work is well written and described throughout. I only have a few comments for the authors to consider.

1) You use the ATPase assays throughout (Fig 1e, 3d, 4ceg, S2, S5f, S7d, S8a, S8b, to follow the loading and activation process and you use terminology like "reduction" or "elevated" yet no statistical tests are used to be certain of these differences.

2) For the pull down assays in Figure, it would have been nice to have the input prior to any L or H salt washes to qualitatively compare relative intensities.

3) "d" in Figure 5 is missing in the figure itself.

4) Need to define the # label within the Figure S10 in the figure legend. I assume this to mean phosphorylation.

5) Last sentence of Results. "series" is misspelled as "sereies"

Version 1:

Reviewer comments:

Reviewer #1

(Remarks to the Author)

The authors have addressed the comments satisfactorily. This work is elegant and provides important mechanistic insights into the role of MCM2-7 ring closure in triggering Mcm4 ATP hydrolysis.

Recommend the following minor changes:

Supplementary 1 legend: fix typo "salt-table" to "salt-stable"

Supplementary figure 2: The current color scheme makes it difficult to distinguish between the blue and purple bars (there are two purple shades, MCM 2-7 and the pre-RC). Adding a clear key with labels for each color or making it clear in the legend would improve clarity and help readers easily identify what each bar represents. What is the purpose of keeping the MCM 2-7 bar as it is not used in any of the calculations?

The key in Supplementary Fig 14 looks good.

Figure 4E: make the 6-WB bar a cyan color as represented in Fig 4A for MCM6 instead of blue which represents MCM3

Supplementary Figure 12: In the results section describing this figure, please specify Supplementary Figure 12"a", not just Supplementary Figure 12 followed by the lanes as both 12a and 12b have the same numbered lanes.

The authors note that We have now quantified the suggested data using band densitometry (n=3): I don't see this data for many figures. For example, Figure S1d, S1e, S1f (to name a few), where is the quantification? It would be helpful to include it.

Reviewer #2

(Remarks to the Author)

This revised manuscript by Faull and colleagues has addressed my major points of concern from the first submission. The illustration of the fit of models to EM maps, and acknowledgement of the limited ability to resolve side chains in the closed OCCM structure brings clarity to the analysis and how specific mutants were designed. The explicit discussion of the Mcm4 mechanism alongside the previously published Cdc6 proofreading mechanism is also an improvement.

The differences between deleting the Mcm5 WHD and mutating it remain puzzling but I accept that resolving these differences is beyond the scope of the current manuscript.

Overall, I would now support publication in Nature Communications.

Reviewer #3

(Remarks to the Author)

The authors have responded well to concerns from all reviewers. Specifically addressing statistical significance in the measured ATPase values and loading efficiencies in various salt washes. The inclusion of additional data regarding the electron density and maps for the WHDs shows more clearly how this 6.1 angstrom structure can provide insights on an unstable (and short lived) intermediate in the loading process. This provides a good mechanism for the establishment of the MO complex and ejection of Cdt1 and is an important advance for the field.

Point-by-point response to review comments

Reviewer #1 (Remarks to the Author):

The formation of the pre-replication complex involves ORC/Cdc6-mediated loading of Cdt1/MCM2-7 onto DNA origins. Significant advancements, particularly through cryo-EM structural analyses and in vitro assays using purified proteins by various research groups, including the Li-Speck groups, have greatly enhanced our comprehension of these complexes. Specifically, we now have a comprehensive understanding of how the OCCM intermediate forms and how this transient complex is detectable in the presence of ATPγS. Subsequent to ATP hydrolysis, the release of Cdc6 and Cdt1, along with ORC flipping, leads to the establishment of MCM2-7-ORC (MO), which subsequently loads the second MCM in a head-to-head configuration. However, there are still pertinent inquiries regarding the structural alterations during MCM ring closure and the stimuli for ATP hydrolysis. This work aims to elucidate these critical questions.

Using elegant biochemical and structural approach, the authors demonstrate that the C-terminus of Mcm5 is essential for preRC formation and Mcm4 ATP-hydrolysis induces a structural change in the MCM2-7 complex triggering the release of Cdt1. The authors use advances in cryo-EM data analysis to reveal a remodeled “closed conformation” structure of the OCCM, that provides new insights into the structural changes and regulatory mechanisms behind MCM2-7 ring closure. Further, the authors provide compelling data related to the key roles played by Mcm4 and the mechanism of Mcm2-7 ring closure leading to Mcm4 ATP hydrolysis and ring splitting. These are important observations and fill a significant gap in the existing literature. I recommend the following to improve the presentation of the data.

Many thanks to the reviewer for their thoughtful comments, kind words and suggestions.

1) One of the key observations is the importance of Mcm4 ATP-hydrolysis. I recommend including that in the title.

We have now incorporated this into the title.

2) Some of the data using the invitro purified proteins and their interpretations are based on visual appearance of the complexes in high or low salt. There should be a better way to quantify this data. For example: In the presence of ATP, Mcm2-7-delC2 resulted in mildly reduced complex formation in low salt-washed reactions (Supplementary Fig 6D, lanes 6-7). This data is not compelling, and authors need to provide some method of quantification. Similarly, for Figure 3c However, low salt-washed reactions containing ATP revealed that all three mutants caused a degree of complex instability. This data is not convincing. Also, Thus, the data indicate that Mcm5 WHD mutants are associated with reduced viability, defective.... There is no statistical significance included for the datasets. Another example, is Supplementary Figure 7E, with Mcm5-O3 showing the least effect and Mcm5-WH displaying the greatest stabilization.’ This reviewer is unable to visualize this observation. Can you quantify this?

Thank you for this suggestion. We have now quantified the suggested data using band densitometry (n=3), added graphs showing the quantification and revised the text for non-significant changes.

Yeast growth assays on plates are commonly not quantified. Still, we have changed the summary section to read: "Thus, the data indicate that Mcm5 WHD mutants are associated with reduced cell growth *in vivo*, defective double hexamer formation and significantly increased ATPase activity ($P < 0.0001$)."

3) Fig 1: D. In the lanes showing pre-RC $\Delta 5$, we see that Cdt1 is not completely released/marginal levels of Cdt1 remains, which is slightly higher than its pre-RC WT counterpart after low salt extraction. In the later sections, we do see the direct role of the ATPase activity of MCM4 in Cdt1 release, which is downstream of the role of Mcm C5. Could loss of Mcm5 C5 activity by itself result in the reduction of Cdt1 release?

Thank you for the question. In this context, it is important to consider that many helicase-loading mutants display increased Cdt1. Due to the reduced stability of mutant complex intermediates, new complex assemblies are formed all the time, which is associated with the recruitment of Cdt1- MCM2-7. This explains why more Cdt1 is observed in Fig. 1d (lane 14) than in the WT reaction (lane 8), which forms more stable helicase-loading intermediates. However, the same can be observed for many other helicase-loading mutants. Besides, biochemical and structural data have never implicated C5 in Cdt1 binding. For these reasons, we suggest that MCM2-7- $\Delta C5$ is less likely to have a Cdt1 release defect.

4) Fig 2: The authors hypothesize that electrostatic interactions between DNA and MCM2-7 contribute to ring closure but do not experimentally verify these interactions. This interpretation is speculative without direct experimental evidence. As a follow-up, they could mutate the positively charged residues on Mcm2 and Mcm5 and measure how stable the OCCM is using assays like EMSA. -Closed ring conformation only accounts for 10% of the particle pool suggesting that this conformation might not be representative-however, given the dynamic and transient nature of the OCCM it might not be possible to capture the model at this specific conformation. -Resolution of closed ring model is 6.1Å which is relatively low.

We understand the reviewer's concern. However, the entire complex of ORC-Cdc6 and Cdt1-Mcm2-7 has evolved to bind DNA, with unusually large and extensive interfaces. And even within the central chamber of MCM2-7, there are numerous structural elements/residues that bind DNA. This makes the prospect of mutation and seeing a clear effect daunting and quite impractical. Nevertheless, we attempted to generate the mutant but detected hexamer instability during purification, which prevented us from biochemically characterising the mutant. Accordingly, we revised the text and Fig. 2d.

We fully agree with the reviewer that this open conformation is a minor species of the particle population, and the most representative structure is the one in which the Mcm2/5 gate has partially closed upon DNA entry into the Mcm chamber. Due to the transient nature, the resolution of the EM map of this rather small particle population is limited. However, a higher resolution structure of the open gate OCCM is available to aid the structural interpretation. Further, the changes that the loading intermediate undergoes are largely of rigid body-like movement, making it feasible to interpret the medium-resolution map with confidence.

5) Fig 3: C. In the absence of ATP hydrolysis, the mutations 5-AW and 5-WH show higher OCCM stability compared to WT. Is this because we find higher levels of ATP hydrolysis (hence higher OCCM disassembly)

in the presence of these mutations compared to WT, so absence of ATP hydrolysis stabilizes the OCCM more than it stabilizes WT?

We do not know the underlying reason for the increased OCCM stability in the presence of ATPgS and 5-AW or 5-WH, so we refrain from speculating.

6) For Supplementary Figure 8a: Can the ATP hydrolysis be shown like the data presented in Figure 4c. Instead of normalizing the data, please show the WT and this would be consistent with how the data is plotted in the entire manuscript. Also, for none of the ATP hydrolysis experiments, statistical significance is calculated. Each experiment must be repeated at least thrice and the results plotted with statistics. This will improve the rigor of the presented data.

Thank you for your good suggestions. We have now changed the presentation of the data in Supplementary Fig. 8a and 7d (now Supplementary Fig. 9d and 10a-c) as suggested. We have performed statistical analysis on all ATPase assay data and reported significance where appropriate. We have also changed the appearance of graphs to make the plotted individual values more visible. We have three repeats for all experiments, except a single mutant (MCM2-7-7RF) in one particular assay (Fig. 4g) that we clearly stated in the figure legend due to the absence of a hot lab in the new building we moved to.

7) Supplementary Fig 9: The results section titled “Mcm4 ATP-hydrolysis leads to a structural change in MCM2-7 triggering Cdt1 release” that discusses this figure would be easier to follow if the text includes references to the figure (specifying what section describes 9A or 9B). The text also talks about the comparison between the closed gate OCCM and MO complex structures, but the all the figures are labeled with “open gate OCCM” which is slightly misleading.

Many thanks for the suggestion. The labelling has been corrected to “closed-gate OCCM” and the text has been updated to specify which figure panel is being referred to.

8) The various linkers for Mcm5 C-terminus were generated (Figure S1c). It would be useful to include a control for another MCM subunit for these experiments and test the addition of a linker to its C-terminus (of course guided by their structure).

We understand that the reviewer is concerned about the linker's impact. To clarify, we replaced the naturally occurring linker with another flexible linker, which retained the function, indicating that the linker sequence is not essential. Since we provide a positive control, we consider this a well-controlled experiment.

Since the WHD appears to be functionally more relevant than the linker we have provided additional data on this. We replaced the Mcm5 WHD with that of Mcm4 (the Mcm4 subunit itself was not modified) and a humanised version of WHD5. These results are shown in new Supplementary Fig. 1f. The data indicate that the Mcm4-WHD cannot replace the Mcm5-WHD and that hWHD is less efficient in loading salt-stable double hexamers. We hope that this demonstrates that the role of WHD5 in pre-RC formation is essential and specific.

9) Figure 4g is labeled Figure 5g in the text ...similar as with Mcm4-WB

This has now been corrected.

10) The ORC phosphorylation experiments with the addition of CDK is interesting but doesn't tackle if a specific ORC is involved. This information didn't seem to fit well with rest of the theme of the manuscript.

Thank you for this comment. CDK phosphorylated ORC has been used in the past to monitor complex disassembly as part of a quality control mechanism. We included CDK phosphorylation to ask which ATPase is active for this important quality control mechanism. For this reason, we did not focus on a specific Orc subunit. We have clarified the reason for using CDK phosphorylation in the manuscript, and upon the suggestion of reviewer 2, we have also added another experiment that utilises CDK to assess loading-incompetent complexes. We hope that this helps to rationalise our use of CDK phosphorylated ORC.

11) Please fix typos-an example included, there are a few more: Cdt1 is binding both the N-termial and C-termianl domains of MCM2-7.

These typos, and additional ones, have now been corrected.

12) Incomplete sentences: To if this process is ATP-dependent, we repeated the experiments using the Mcm4-WB mutant.

This sentence has now been completed.

13) Fig 5-Label for 5D is missing on the figure.

This has now been corrected.

Reviewer #2 (Remarks to the Author):

In this manuscript, Faull et al use a combination of cryo-EM, biochemistry and genetics to examine the role of various protein functions and domains during pre-RC assembly. Visualising a new intermediate in the MCM loading process, they suggest that the MCM5 winged helix domain closes the MCM2-7 ring in the OCCM complex, and that failure to do so triggers release of MCM2-7 in a manner that is dependent on the MCM4:MCM7 ATPase site. The focus of the manuscript is an interesting one; the text is well written, and the experiments have been well executed. However, in my opinion there are some key aspects of the manuscript that currently preclude publication in Nature Communications.

Many thanks to the reviewer for their thoughtful comments, kind words and suggestions.

Major points.

1. More detail of the processing steps involved in obtaining the new closed ring OCCM structure should be provided. In particular, a supplementary figure describing the processing steps, particle numbers involved etc should be added to the manuscript.

Thanks for the suggestions. We have included the processing steps in the revised manuscript (revised Supplementary Fig. 3).

2. Related to this, while I can appreciate that the intermediate nature of the closed ring OCCM structure may limit the resolution that can be obtained, 6.1 Å is likely to limit the conclusions that can be drawn from the structure. More detail is required around this. For example, there is no description at all in the text or figures of how well the density in their final map can be fitted to a model of the C-terminus of MCM5. Is the density unambiguously the MCM5 WHD or could another domain also be fitted here? This is a crucial point. The same applies to the MCM2 residues shown in figure 2d – can sidechain densities for these residues be observed in the map? No fit of the model to the map is shown. If not, this should be removed.

We have incorporated the model-map fitting into the manuscript (Supplementary Fig. 4a-c). Due to the low population of the “closed-gate” conformation, the number of particles is limited, which is the primary reason for the low-resolution map. In the cryo-EM map of gate closed OCCM, the side chain density is not visible at the current resolution. However, due to the availability of the higher resolution structure of the gate-open OCCM, and the fact that the conformational changes from open-to-close transition is largely rigid-body movement, we can dock the available structure into the gate-closed EM map, and to interpret the low-resolution structure with some confidence. We were only noting the presence of certain positively charged residues around the DNA, and we didn’t specify their distances to DNA and whether they are definitely interacting. However, in light of the reviewer’s concerns, we have removed the labelling of specific residues in Fig. 2d and revised the description of the protein-DNA interactions in the manuscript.

3. While I appreciate that this may be challenging, no validation of the MCM5 WHD position, or description of any attempts to validate its position are provided. Cross linking mass spectrometry could be one way to do this.

Despite the modest resolution, the position of MCM5 WHD is quite certain, as demonstrated by newly added panel c in Supplementary Fig. 4. There are only 5 WH domains in MCM2-7, and every one of these domains is accounted for. We can be certain of the assignment due to the resolution we reached is sufficient to resolve the alpha helices; there is a high-resolution structure of the related conformation of OCCM (gate open); and then changes from gate open to close is largely rigid-body based movements.

4. They describe several point mutations in MCM5 that they state are at the interface of MCM5 WHD and either the AAA+ domain or the ORC3 interface. No evidence is shown for this. A figure showing the electron density map and model and the positions of the amino acids that are mutated should be shown. Again, how sure are the authors on the location of these residues given the resolution of the structure?

Thank you for your comment. Although the side chain of the residue in Mcm5 WHD is not visible, it fits well into the density (new Supplementary Fig. 4c). The previously determined high-resolution structure of Mcm5 assists us in docking the Mcm5 WHD into the cryo-EM map as a rigid body. Despite the low resolution and lack of precise side chain position information, we can still predict which residues might facilitate the interaction between Orc3 and Mcm5. We decided to mutate multiple residues, most of which are conserved across species (Supplementary Fig. 9b), as opposed to using single point mutations and used charge swapping to ensure disruption of the predicted interfaces. The positions of all mutants are shown in Fig. 3a.

5. Perhaps related to the previous point, the biochemical impact of mutating the MCM5 WHD is different from deleting it. For example, MCM5 WHD deletion does not cause an increase in 'pre-RC induced ATP hydrolysis' (figure 1), whereas all mutations they study in the MCM5 WHD do (figure 3). This does not fit with their model that any failure to close the 2/5 gate triggers ATP hydrolysis by MCM4. How do the authors rationalise this? The MCM5 Δ WHD should be analysed alongside the MCM5 WHD point mutations in figure 3 as a point of reference.

Thank you for the comment and question. We note that the deletion of C5 drastically impacted complex stability but did not result in a further increase of pre-RC-induced ATPase activity that was observed in C5 point mutants. We suggest that C5 has an additional function in stabilising complex intermediates. Interestingly, the WHD of Mcm4 can bypass this severe complex instability of C5 deletion (compare Fig. 1D and Supplementary Fig 1F). Thus, C5 may have a sequence-independent structural role. Considering that C5 sits within the central pore of the Cdt1-MCM2-7 complex and the pre-insertion OCCM (a complex arrested before DNA insertion), C5 may also function in stabilising the MCM2-7 ring during the initial stages of OCCM formation or regulate DNA insertion. Since these steps occur before ATP-hydrolysis, they may result in ATPase-independent instabilities and complex disassembly.

Regarding the second point, we have included a new supplementary figure that shows the induced ORC/Cdc6 and MCM2-7/Cdt1 values for all mutants used in the manuscript (Supplementary Fig. 14). We have displayed MCM2-7- Δ C5 next to the Mcm5 mutants and used the same scale for all graphs to allow easy comparison. We prefer this, as it allows comparison of all data. To further highlight the induction of ATP hydrolysis caused by mutating C5, we have included Supplementary Fig. 9d, which shows the percentage change in ATP hydrolysis for MCM2-7- Δ C5 and the WHD mutants compared to their WT control. Mcm5-O3 and Mcm5-WH have a significant increase in ATP hydrolysis compared to MCM2-7- Δ C5.

6. Given that, as referenced in the paper, the Diffley lab previously described an ORC-Cdc6-dependent quality control mechanism, the authors should compare the role of Cdc6 and Mcm4 ATPase in removing loading-incompetent MCM complexes. Are they both required to remove the same loading-incompetent complexes?

Thank you for this suggestion. We performed a pre-RC assay on CDK-phosphorylated ORC, which is not competent for pre-RC formation and triggers quality control (PMID: 23474987, PMID: 23603117) with Mcm4-WB and Cdc6 E224Q in the presence of ATP and ATP γ S (Supplementary Fig. 12 b and c). More recent single-molecule and cryo-EM experiments by the Bell and Bleichert labs (PMID: 37428921, PMID: 35217664) have indicated that CDK phosphorylation of ORC affects productive docking of MCM2-7 and is associated with release of the Cdt1-MCM2-7. This event occurs before DNA insertion. Our analysis showed greater stabilisation levels in the presence of Cdc6 E224Q than Mcm4-WB (Supplementary Fig. 12). Thus, the data suggest that Cdc6 ATPase functions in quality control-dependent removal of Cdt1-MCM2-7 before

DNA insertion, while Mcm4 ATPase is triggered due to structural changes in the MCM2-7 ring following DNA insertion. Accordingly, we revised the manuscript.

Minor points.

1. There are a number of typos throughout the manuscript.

Thank you for highlighting this, we have checked and corrected multiple typos.

2. For the 'pre-RC induced ATP hydrolysis', the supplementary figure helps to understand where this figure derives from. However, the individual values going into this final figure are important and should be included in the main text/figures for each mutant analysed.

Thank you for this comment. We have included graphs for all mutants in Supplementary Fig. 14. These graphs have been plotted using the same scale to allow for easy comparison. The individual values used to plot graphs will be available as a supplementary table.

Reviewer #3 (Remarks to the Author):

Review of MCM2-7 ring closure involves the Mcm5 C-terminus and triggers MCM2-7 ATP hydrolysis by Faull et al. for Nature Communications. In this manuscript, the authors have added to our knowledge of MCM2-7 loading and activation by revealing three further aspects of this process. 1) They show that the winged helix domain within the CTE of MCM5 is influential in interacting with ORC3 to stabilize a closed OCCM complex, 2) that the ATPase activity of MCM4 is necessary for closing the MCM2-5 interface prior to C5 movement, and that 3) Cdt1 is ejected during this process. They also suggest that any defect in MCM2-7 ring closure would disassemble the entire complex with MCM4 ATPase activity providing a failsafe for properly loaded and closed MCM2-7 complexes. The work is well written and described throughout. I only have a few comments for the authors to consider.

1) You use the ATPase assays throughout (Fig 1e, 3d, 4ceg, S2, S5f, S7d, S8a, S8b, to follow the loading and activation process and you use terminology like "reduction" or "elevated" yet no statistical tests are used to be certain of these differences.

Thank you for this observation. We have now analysed the data, indicated which comparisons are statistically significant and modified the text accordingly.

2) For the pull-down assays in Figure, it would have been nice to have the input prior to any L or H salt washes to qualitatively compare relative intensities.

We have now added inputs for the pulldown assays.

3) "d" in Figure 5 is missing in the figure itself.

This has now been corrected.

4) Need to define the # label within the Figure S10 in the figure legend. I assume this to mean phosphorylation.

The # in Supplementary Fig. 10 does indicate phosphorylation, this denotation has been added to the figure legend.

5) Last sentence of Results. "series" is misspelled as "sereies"

This has now been corrected.

REVIEWERS' COMMENT

Reviewer #1 (Remarks to the Author):

The authors have addressed the comments satisfactorily. This work is elegant and provides important mechanistic insights into the role of MCM2-7 ring closure in triggering Mcm4 ATP hydrolysis.

We thank reviewer #1 one for the careful reading of our work and positive comments in support of publication.

Recommend the following minor changes:

Supplementary 1 legend: fix typo “salt-table” to “salt-stable”

This has been corrected.

Supplementary figure 2: The current color scheme makes it difficult to distinguish between the blue and purple bars (there are two purple shades, MCM 2-7 and the pre-RC). Adding a clear key with labels for each color or making it clear in the legend would improve clarity and help readers easily identify what each bar represents. What is the purpose of keeping the MCM 2-7 bar as it is not used in any of the calculations?

The key in Supplementary Fig 14 looks good.

Supplementary Fig.2 has now been amended to change the colour scheme, using Supplementary Fig. 14 (now Supp. Fig. 15) as a guide and we have added a key. We have removed the MCM2-7 bar as we agree that we have not reported this value for the MCM2-7 mutants.

Figure 4E: make the 6-WB bar a cyan color as represented in Fig 4A for MCM6 instead of blue which represents MCM3

This has been amended.

Supplementary Figure 12: In the results section describing this figure, please specify Supplementary Figure 12“a”, not just Supplementary Figure 12 followed by the lanes as both 12a and 12b have the same numbered lanes.

We have now clarified which panel of the figure that we are referring to.

The authors note that We have now quantified the suggested data using band densitometry (n=3): I don't see this data for many figures. For example, Figure S1d, S1e, S1f (to name a few), where is the quantification? It would be helpful to include it.

As requested by the reviewers, we included densitometry of specified gels in our resubmission. This focussed on experiments, where the differences were not black and white. We refrain from

quantifying other experiments for the following reasons: The remaining data are black and white and therefore quantification will not improve the conclusion. It is not common to have these types of experiments quantified.

<https://www.nature.com/articles/s41467-024-52408-0>;

<https://www.sciencedirect.com/science/article/pii/S109727652300761X?via%3Dihub>

<https://www.nature.com/articles/s41594-021-00698-z>

Reviewer #2 (Remarks to the Author):

This revised manuscript by Faull and colleagues has addressed my major points of concern from the first submission. The illustration of the fit of models to EM maps, and acknowledgement of the limited ability to resolve side chains in the closed OCCM structure brings clarity to the analysis and how specific mutants were designed. The explicit discussion of the Mcm4 mechanism alongside the previously published Cdc6 proofreading mechanism is also an improvement.

The differences between deleting the Mcm5 WHD and mutating it remain puzzling but I accept that resolving these differences is beyond the scope of the current manuscript.

Overall, I would now support publication in Nature Communications.

Thank you to reviewer #2 for their comments and helping in revising the manuscript. We accept that the difference in activity of the Mcm5 deletions versus mutation is puzzling, and we thank the reviewer for acknowledging that this is beyond the scope of our work.

Reviewer #3 (Remarks to the Author):

The authors have responded well to concerns from all reviewers. Specifically addressing statistical significance in the measured ATPase values and loading efficiencies in various salt washes. The inclusion of additional data regarding the electron density and maps for the WHDs shows more clearly how this 6.1 angstrom structure can provide insights on an unstable (and short lived) intermediate in the loading process. This provides a good mechanism for the establishment of the MO complex and ejection of Cdt1 and is an important advance for the field.

We appreciate reviewer #3's kind comments and previous suggestions for the improvement of the manuscript.